# LATENT-TO-OBSERVABLE SCORE CORRECTION FOR PROBABILISTIC TIME SERIES IMPUTATION

## ABSTRACT

Missing data remains a key challenge in multivariate time series modeling, often degrading downstream performance. Recent score-based generative models show strong potential for high-quality imputations, yet most ignore original missing data during training, since ground truth is unavailable, resulting in biased score estimation. We theoretically analyze the effect of missingness on score-based modeling under the denoising diffusion probabilistic model (DDPM) framework. Our findings reveal that ignoring original missing patterns—especially under high missing rates or strong inter-variable correlations—can significantly distort the learned score function even at non-missing points. To overcome this, we propose the Hierarchical Score-Based Generative Model (HSGM) for probabilistic time series imputation. HSGM integrates latent-space and observation-space diffusion in a layer-wise refinement framework grounded in the chain rule of probability. A pretrained Variational Autoencoder (VAE) with normalizing flows captures complex latent distributions, while a continuous-time variational diffusion (VPSDE) operates in latent space. A cross-attention mechanism between the original and denoised latent states enhances the fidelity and resolution of the generative outputs, while an observation-space diffusion module further refines the final imputations. Experiments on four benchmark datasets show that HSGM achieves the best accurate imputations with tighter uncertainty estimates than existing methods, while effectively correcting score function bias, establishing a new state of the art in time series imputation.

## 1 INTRODUCTION

Missing data in multivariate time series (MTS) is ubiquitous during data collection, arising from factors such as sensor unreliability and network instability Wang et al. (2024); Miao et al. (2022). Such missingness can significantly degrade the performance of data-driven models in downstream tasks, making multivariate time series imputation (MTSI) a crucial solution Jin et al. (2024); Fang & Wang (2020). Recently, deep learning—particularly score-based models—has achieved remarkable progress in MTSI.

Most imputation approaches based on score-based models simulate missing masks and values to estimate the (conditional) score function, often ignoring the original missing entries due to the lack of ground truth Tashiro et al. (2021); Yang et al. (2024). Common heuristics, such as zero- or mean-imputation, assume that original missing values are independent of both observed and simulated data—an assumption rarely valid in real-world datasets. In practice, the original missing values often exhibit strong temporal or spatiotemporal correlations Cao et al. (2025); Wang et al. (2025); Yuan & Qiao, and ignoring them can bias score estimation. This is particularly problematic in domains like healthcare, where missing rates can be around 80%, making the original missing data too prevalent to disregard Xu et al. (2023); Dai et al. (2024); Liu et al. (2023a). A central challenge in score-based imputation is mitigating bias in the score-matching objective. Popular approaches, including MissDiff Ouyang et al. (2023), mask the conditional score-based function during denoising score matching but typically ignore dependencies between observed and original missing data, leading to suboptimal score estimates. To address this, Givens et al. Givens et al. (2025) proposed importance weighting (IW) and variational approximations of the true score. While IW mitigates distributional shifts by reweighting samples, it can induce high variance and unstable gradients when the weights

are poorly estimated. Variational methods, such as the Marginal Variational (Marg-Var) approach based on the Expectation-Maximization (EM) algorithm, provide improved stability but often rely on manual feature engineering, limiting end-to-end training. DiffPuter Zhang et al. (2025) integrates diffusion models with EM to tackle missing data imputation, iteratively learning the joint distribution of observed and missing values while performing conditional sampling. These approaches remain computationally demanding due to repeated score approximations at each diffusion step.

In this paper, we analyze the original missing effect in learning the score-based function from a mathematical perspective under the denoising diffusion probabilistic model (DDPM) framework. To explicitly account for original missing values, we propose a Hierarchical Score-Based Generative Model (HSGM) for probabilistic time series imputation. Inspired by the layer-wise refinement paradigm of multilayer perceptrons (MLPs), HSGM leverages both latent and observation score-based diffusion models to capture the latent distribution of the dataset. The original missing values are reconstructed via the latent diffusion process and a Variational Autoencoder (VAE) decoder without requiring ground-truth values, while the observation diffusion layer conditions on these reconstructed values to learn a more accurate score-based function, thereby producing improved imputations in the observation space. The main contributions of our work are as follows:

1. **Theoretical analysis of bias in score-based functions under the observation DDPM framework**. We rigorously show that ignoring original missing data—especially under high missing rates or strong inter-variable correlations—can lead to substantial bias in the learned score function, even for observed points.

2. **Hierarchical latent-to-observation diffusion framework**. Inspired by the layer-wise refinement paradigm of MLP, we theoretically integrate latent-space and observation-space diffusion in a layer-wise refinement framework grounded in the chain rule of probability. This approach corrects the bias in score estimation induced by original missing values, enabling accurate modeling of complex, non-Gaussian data distributions while adaptively handling original missing data without requiring ground-truth supervision during training.

3. **Cross-attention and continuous latent diffusion for high-fidelity imputation**. To balance generative flexibility with reconstruction accuracy, we introduce cross-attention mechanisms between original and denoised latent variables, along with continuous latent diffusion implemented via Ordinary Differential Equation (ODE) sampling. These components jointly guide the generative process, yielding high-fidelity and accurate imputed outputs.

## 2 RELATED WORK

**Variational Generative Models:** VAEs Fortuin et al.; Lee et al. (2022); Kingma & Welling represent one of the earliest and most widely adopted generative approaches for multivariate time series imputation (MTSI). By introducing probabilistic latent variables, VAEs capture the underlying data distribution, encode meaningful variations, and explicitly model uncertainty Vahdat & Kautz (2020). This probabilistic formulation offers a principled and interpretable alternative to deterministic models Zhao et al. (2024). Furthermore, VAEs operate naturally in an unsupervised learning paradigm, making them well-suited for real-world scenarios where the original missing phenomenon prevents access to ground-truth labels.

**Score-based Models in Observation Space:** Score-based models have recently attracted considerable attention for time series imputation due to their theoretical rigor and ability to generate high-quality outputs Yang et al. (2023). CSDI Tashiro et al. (2021) formulates imputation as a conditional diffusion process, using a transformer to capture inter-feature dependencies. PriSTI Liu et al. (2023b) extends this by incorporating conditional features to model temporal and spatial correlations. MTSCI Zhou et al. (2024) enforces intra- and inter-consistency via masking and conditional mixup, while MIDM Wang et al. (2023) re-derives the ELBO to explicitly model consistency between observed and missing values through redesigned noise processes. SADI Dai et al. (2024) leverages cross-time, cross-feature, and cross-patient information for temporal EHR imputation, and FGTI Yang et al. (2024) emphasizes residual components with high-frequency filtering, integrating frequency-domain insights with deep representations. Collectively, these methods highlight the importance of consistency, structured information, and frequency-aware modeling for accurate imputation. DiffPuter Zhang et al. (2025) combines diffusion models with the Expectation-Maximization algorithm to address missing data imputation. It iteratively learns the joint distribution of observed

and missing values and performs conditional sampling. Furthermore, in observation-based Score-based model settings, the original missingness is typically assumed to be independent of both observed and original missing data. In practice, original missing entries are often replaced with zeros or mean values during training, which introduces substantial bias into the learned score function and undermines the model's ability to faithfully capture uncertainty.

**Latent Score-based Models:** Recent research has explored integrating VAEs with score-based models to enhance imputation quality Zhang et al. (2024). For instance, LSSDM Liang et al. (2025) adopts a two-stage approach, first imputing originally missing data with a VAE, followed by a diffusion process. Nevertheless, the imputation remains constrained by typical VAE limitations, such as blurry reconstructions and limited capacity to capture complex distributions. Inspired by Stable Diffusion Rombach et al. (2022), latent diffusion has shown success in generating high-resolution outputs in vision tasks Croitoru et al. (2023); Ma et al. (2025); Corneanu et al. (2024). LDT Feng et al. (2024) features a symmetric statistics-aware autoencoder for learning time series latents and a diffusion-based conditional generator for flexible future prediction. Applying this paradigm to imputation, however, presents unique challenges: due to the nonlinear mapping between latent and observation spaces, small perturbations in latent variables can lead to disproportionately large bias in reconstructed data, compromising robustness. Conditional guidance mechanisms partially mitigate this issue by aligning the latent diffusion with observed data distributions Ni et al. (2023); Van Gansbeke & De Brabandere (2024), yet they cannot fully eliminate irrelevant noise introduced during latent sampling, necessitating further refinement of the outputs. Moreover, balancing generative flexibility and reconstruction fidelity in latent diffusion remains an open problem. VA-VAE Yao et al. (2025) proposes a Vision Foundation model alignment loss, combining marginal cosine similarity and distance matrix losses in the latent space. However, it only considers complete datasets and neglects the impact of missing values within the latent representations.

## 3 BACKGROUND

### 3.1 PRELIMINARY

Let $\mathbf{X}_0 \in \mathbb{R}^{N \times F}$ denote the complete dataset and $\mathbf{M} \in \mathbb{R}^{N \times F}$ the missing mask, where $\mathbf{M}_{i,j} = 0$ indicates that the $j$-th sensor at time $i$ is missing, and $\mathbf{M}_{i,j} = 1$ otherwise. Similarly, $\mathbf{X}_{0(ij)}$ or $x_{0(ij)}$ denotes the $(i, j)$-th entry of $\mathbf{X}_0$ in this study. Missing values are categorized as original missing data (ground-truth unavailable) and simulated missing data (used for training and evaluation), leading to $\mathbf{X}_0^{Or}$ and $\mathbf{X}_0^{Ta}$, with masks $\mathbf{M}^{Or}$ and $\mathbf{M}^{Ta}$, respectively. Thus, $\mathbf{X}_0^{Ta} = \mathbf{X}_0 \odot (1 - \mathbf{M}^{Ta})$, $\mathbf{X}_0^{Or} = \mathbf{X}_0 \odot (1 - \mathbf{M}^{Or})$ and $\mathbf{X}_0^{\overline{Or}} = \mathbf{X}_0 \odot \mathbf{M}^{Or}$. Additionally, simulated and original missing masks do not overlap, $(\mathbf{M}^{Or} = 0) \cap (\mathbf{M}^{Ta} = 0) = \varnothing$. The conditional observed data is defined as $\mathbf{X}_0^{Co} = \mathbf{X}_0 - \mathbf{X}_0^{Ta} - \mathbf{X}_0^{Or} = \mathbf{X}_0 \odot \mathbf{M}^{Or} \odot \mathbf{M}^{Ta}$. Visualization of how the matrix of the available data is created is provided in Appendix A.1.

### 3.2 REVIEW OF DDPM MODEL

Diffusion models can be formulated within the framework of a general stochastic differential equation (SDE). One of the SDE diffusion methods is DDPM Ho et al. (2020), a class of generative score-based models that learn to reverse a gradual noising process. Generative models aim to learn data distributions and generate realistic samples. DDPMs are a recent class that generate data by reversing a diffusion process. We define a sequence of latent variables $\mathbf{X}_0, \mathbf{X}_1, \ldots, \mathbf{X}_T$, where $\mathbf{X}_0 \sim q(\mathbf{X}_0)$ is the data and DDPM as:

$$q(\mathbf{X}_t \mid \mathbf{X}_{t-1}) = \mathcal{N}\left(\mathbf{X}_t; \sqrt{1 - \beta_t}\,\mathbf{X}_{t-1}, \beta_t \mathbf{I}\right), \tag{1}$$

with a variance schedule $\beta_1, \ldots, \beta_T$. This formulation leads to the conditional distribution:

$$q(\mathbf{X}_t | \mathbf{X}_0) = \mathcal{N}(\mathbf{X}_t; \sqrt{\bar{\alpha}_t}\mathbf{X}_0, (1 - \bar{\alpha}_t)\mathbf{I}), \tag{2}$$

where $\bar{\alpha}_t := \prod_{j=1}^t (1 - \beta_j)$. The model is trained by optimizing a re-weighted evidence lower bound (ELBO) Song et al.:

$$\theta^* = \arg\min_\theta \sum_{i=1}^T (1 - \bar{\alpha}_t)\mathbb{E}_{p_{\text{data}}(\mathbf{X}_0)}\mathbb{E}_{q(\mathbf{X}_t|\mathbf{X}_0)}\left[\|\nabla_{\mathbf{X}_t}\log p_\theta(\mathbf{X}_t) - \nabla_{\mathbf{X}_t}\log q(\mathbf{X}_t|\mathbf{X}_0)\|^2\right]. \tag{3}$$

We further denote $\mathbf{s}_\theta(\mathbf{X}_t, t) = \nabla_{\mathbf{X}_t} \log p_\theta(\mathbf{X}_t)$ as the parameteric score function. Through reparameterization and after ignoring the constant term of $\sqrt{1 - \bar{\alpha}_t}$, the simplified objective function can be written as:

$$\mathcal{L} = \mathbb{E}_{\mathbf{X}_0, \epsilon, t} \left[ \| \epsilon - \epsilon_\theta(\mathbf{X}_t, t) \|^2 \right], \tag{4}$$

where $\epsilon \sim \mathcal{N}(\mathbf{0}, \mathbf{I})$ and $\mathbf{X}_t = \sqrt{\bar{\alpha}_t} \mathbf{X}_0 + \sqrt{1 - \bar{\alpha}_t} \, \epsilon$. After obtaining the optimal model $\mathbf{s}_{\theta*}(\mathbf{X}_t, t)$, new samples can be generated through the following reverse process:

$$\mathbf{X}_{t-1} = \frac{1}{\sqrt{1 - \beta_t}} (\mathbf{X}_t + \beta_t \mathbf{s}_{\theta*}(\mathbf{X}_t, t)) + \sqrt{\beta_t} \epsilon, \tag{5}$$

where $\epsilon \sim \mathcal{N}(\mathbf{0}, \mathbf{I})$. DDPMs provide stable training and high-quality generation, outperforming many Generative Adversarial Networks(GANs) in sample quality.

## 4 BIAS ANALYSIS OF SCORE-BASED DIFFUSION WITH MISSING EFFECT

**Proposition 4.1** (Bias under Independent Assumption in DDPM Setting). *Consider a data matrix* $\mathbf{X}_0$ *in the DDPM score-based diffusion setting, where each entry is independent. Assume that the score-based function* $\mathbf{s}_{ij}(\mathbf{X}_t, t)$ *is differentiable at* $\mathbf{X}_t^{\overline{Or}}$. *Then the bias of the score-based function at time step* $t$ *is:*

$$\mathbf{s}_{bias(ij)} = \mathbf{s}_{ij}(\mathbf{X}_t, t) - \mathbf{s}_{ij}(\mathbf{X}_t^{\overline{Or}}, t) = -\frac{\sqrt{\bar{\alpha}_t} x_{0(ij)}}{1 - \bar{\alpha}_t}. \tag{6}$$

Proof is provided in Appendix A.2.

We plot the relative approximation bias under the setting of total time steps $T = 200$, $\beta_0 = 0.02$, and $\beta_T = 0.5$. As shown in Fig. 1a, the bias of the score function is relatively large in the early steps but gradually diminishes as $t$ increases.

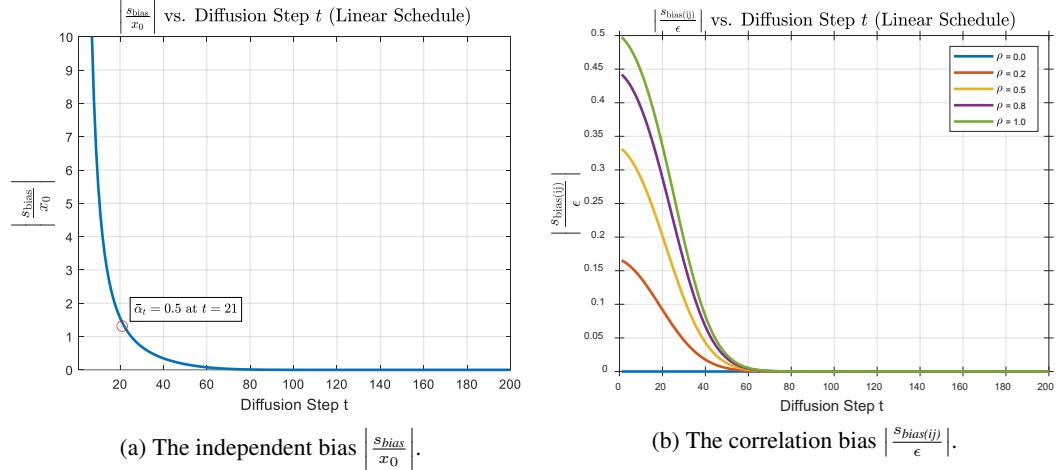

(a) The independent bias $\left| \frac{s_{bias}}{x_0} \right|$.

(b) The correlation bias $\left| \frac{s_{bias(ij)}}{\epsilon} \right|$.

Figure 1: Bias plot under different assumptions.

**Corollary 4.1** (Expected Cumulative bias). *Under the assumptions of Proposition 4.1, the expected cumulative bias over* $T$ *diffusion steps is proportional to the original missing rate* $p^{Or} = \frac{\sum_{i=0}^{N} \sum_{j=0}^{F} (1 - \mathbf{M}_{ij}^{Or})}{NF}$:

$$\mathbb{E}\left[ \sum_{t=1}^{T} \mathbf{s}_{bias} \right] = p^{Or} \sum_{t=1}^{T} \sum_{i,j \in Or} \left[ -\sqrt{\bar{\alpha}_{t-1}} \frac{1 - \alpha_t}{1 - \bar{\alpha}_t} x_{0(ij)} \right]. \tag{7}$$

Proof is provided in Appendix A.3.

From Corollary 4.1, the bias is proportional to the original missing rate. Consequently, when the missing rate is high, the effect of missingness cannot be neglected.

**Proposition 4.2** (Bias under Correlated Gaussian Data in DDPM Setting). *Consider two correlated points $x_{0(ij)}$ and $x_{0(kl)}$ in the DDPM score-based diffusion setting, which follow a joint Gaussian distribution with correlation coefficient $\rho$ and standard deviation $\sigma$. If $x_{0(kl)}$ is originally missing and replaced by zero, then the bias on the observed point $x_{0(ij)}$ is:*

$$\mathbf{s}_{bias(ij)} = s_{ij}(x_{t(ij)}, t) - s_{ij}(x_{t(ij)}^{\overline{Or}}, t)$$

$$= \frac{1}{D_t} \left[ \left(\bar{\alpha}_t \sigma_{kl}^2 + (1 - \bar{\alpha}_t)\right)\left(x_{t(ij)} - \sqrt{\bar{\alpha}_t}\mu_{ij}\right) - \bar{\alpha}_t \rho \sigma_{ij} \sigma_{kl}\left(x_{t(kl)} - \sqrt{\bar{\alpha}_t}\mu_{kl}\right) \right] - \frac{x_{t(ij)} - \sqrt{\bar{\alpha}_t}\mu_{ij}}{\bar{\alpha}_t \sigma_{ij}^2 + (1 - \bar{\alpha}_t)}.$$
(8)

*where $D_t = \left(\bar{\alpha}_t \sigma_{ij}^2 + (1 - \bar{\alpha}_t)\right)\left(\bar{\alpha}_t \sigma_{kl}^2 + (1 - \bar{\alpha}_t)\right) - (\bar{\alpha}_t \rho \sigma_{ij} \sigma_{kl})^2$. This shows that the bias on observed points increases with the correlation coefficient $\rho$.* Proof is provided in Appendix A.4.

Given $\sigma_{ij} = 1$ and $\sigma_{kl} = 1$, we plot $\left|\frac{s_{bias(ij)}}{\epsilon}\right| = \frac{|\bar{\alpha}_t \rho|}{1 + \bar{\alpha}_t \rho}$. According to Eq. 8 with varying $\rho$, under the same setting as Proposition 4.1. As shown in Fig. 1b, the bias decreases as the diffusion step $t$ increases, but becomes more pronounced as the correlation coefficient $\rho$ grows at the early steps. In practice, time-series sensors are often highly correlated with both their temporal and spatial neighbors. Therefore, even when training with simulated missing values for which ground-truth data are available, the effects of the original missing values—intrinsic to the dataset—cannot be ignored.

## 5 METHODOLOGY

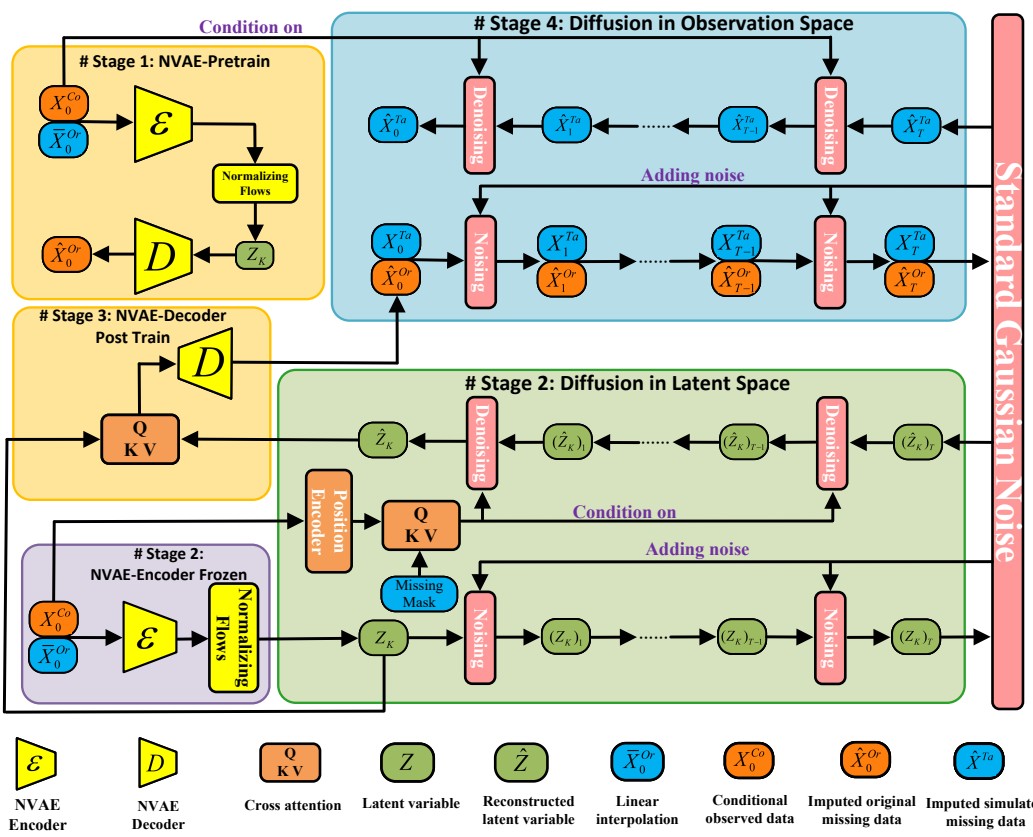

Figure 2: **Stage 1:** NVAE Vahdat & Kautz (2020) with normalizing flows is pretrained to obtain non-Gaussian latent variables $\mathbf{Z}_K$ from observed and interpolated data. **Stage 2:** $\mathbf{Z}_K$ is refined via a continuous latent diffusion model with cross-attention over the missing mask and position encoding (encoder is frozen). **Stage 3:** reconstructed $\hat{\mathbf{Z}}_K$ is aligned with $\mathbf{Z}_K$ through cross-attention, and the NVAE decoder is fine-tuned to produce $\hat{\mathbf{X}}_0^{Or}$. **Stage 4:** a final diffusion step in the observation space yields the imputed data $\hat{\mathbf{X}}_0^{Ta}$.

## 5.1 Score correction by latent score-based generative models

We have previously shown that correlations such as $\mathbf{X}_0^{Co} \not\perp\!\!\!\perp \mathbf{X}_0^{Ta} \not\perp\!\!\!\perp \mathbf{X}_0^{Or}$ can induce substantial estimation errors. To mitigate this bias, we propose a latent score-based generative model that explicitly captures these dependencies. A central challenge is correcting the bias in the score function, as the ground-truth values of the originally missing data are inherently unobserved. To tackle this, we exploit the chain rule of probability to decompose the learning objective into two tractable terms:

$$-\log p(\mathbf{X}_0^{Ta}, \mathbf{X}_0^{Or}|\mathbf{X}_0^{Co}) \approx -\log \left[ \int p(\mathbf{X}_0^{Ta}|\mathbf{X}_0^{Co}, \hat{\mathbf{X}}_0^{Or}) p(\hat{\mathbf{X}}_0^{Or}|\mathbf{X}_0^{Co}, \mathbf{Z}_0) p(\mathbf{Z}_0|\mathbf{X}_0^{Co}) d\mathbf{Z}_0 \right]$$

$$= \underbrace{-\log p(\mathbf{X}_0^{Ta}|\mathbf{X}_0^{Co}, \hat{\mathbf{X}}_0^{Or})}_{\mathcal{L}_1 (\textit{Observation term})} \underbrace{- \log \int p(\hat{\mathbf{X}}_0^{Or}|\mathbf{X}_0^{Co}, \mathbf{Z}_0) p(\mathbf{Z}_0|\mathbf{X}_0^{Co}) d\mathbf{Z}_0}_{\mathcal{L}_2 (\textit{Latent term})}. \quad (9)$$

For the observation term $\mathcal{L}_1$, we adopt the conditional score-based diffusion model (CSDI) Tashiro et al. (2021) to compute the conditional probability, conditioned on the reconstructed outputs $\hat{\mathbf{X}}_0^{Or}$ obtained from the latent term $\mathcal{L}_2$. Details are provided in Appendix A.5. For the latent term $\mathcal{L}_2$, we employ normalizing flows with $K$ layers to transform the objective into the following form:

$$\mathcal{L}_2 = \underbrace{\mathbb{E}_q \left[ -\log p(\hat{\mathbf{X}}_0^{Or}|\mathbf{X}_0^{Co}, \mathbf{Z}_K) \right]}_{\text{reconstruction term}} + \underbrace{\mathbb{E}_q \left[ \log q(\mathbf{Z}_0|\bar{\mathbf{X}}_0^{Or}, \mathbf{X}_0^{Co}) - \sum_{k=1}^{K} \log \left| \det \frac{\partial f_k}{\partial \mathbf{Z}_{k-1}} \right| \right]}_{\text{negative enconder entropy}} + \underbrace{\mathbb{E}_q \left[ -\log p(\mathbf{Z}_K|\mathbf{X}_0^{Co}) \right]}_{\text{cross entropy}}$$

$$(10)$$

where $\bar{\mathbf{X}}^{Or}$ denotes the linear interpolation of the original missing data, $\det$ is the determinant, $f$ is the planar flow, and $\mathbf{Z}_K$ is the output of the normalizing flow; see Appendix A.6 for details.

## 5.2 Continuous latent diffusion

In the latent space, for the cross-entropy term, we adopt the variance-preserving SDE (VPSDE), defined as $d\mathbf{z} = -\frac{1}{2}\beta(t)\mathbf{z}dt + \sqrt{\beta(t)}d\mathbf{w}$, where $\beta(t) = \beta_{\text{start}} + (\beta_{\text{end}} - \beta_{\text{start}})t, t \in [0, 1]$. Thus, the forward process can be defined as Song et al.:

$$q((\mathbf{Z}_K)_t|\mathbf{Z}_K) = \mathcal{N}\left((\mathbf{Z}_K)_t; e^{\left(-\frac{1}{2}\beta_{\text{start}}t - \frac{1}{4}(\beta_{\text{end}} - \beta_{\text{start}})t^2\right)}\mathbf{Z}_K, \mathbf{I} - \mathbf{I}e^{\left(-\beta_{\text{start}}t - \frac{1}{2}(\beta_{\text{end}} - \beta_{\text{start}})t^2\right)}\right), \quad t \in [0, 1]$$

$$(11)$$

Following the previous work as LSGM Vahdat et al. (2021), the cross entropy term in the continuous situation can be calculated in an unweighted explicit score matching (ESM) setting as:

$$\mathbb{E}_q \left[ \log p(\mathbf{Z}_K|\mathbf{X}_0^{Co}) \right] = \mathbb{E}_{t \sim \mathcal{U}[0,1]} \left[ \mathbb{E}_{q(\mathbf{Z}_0|\bar{\mathbf{X}}_0^{Or}, \mathbf{X}_0^{Co}), \boldsymbol{\epsilon} \sim \mathcal{N}(0, \mathbf{I})} \left[ \frac{1}{2} ||\boldsymbol{\epsilon} - \boldsymbol{\epsilon}_\theta((\mathbf{Z}_K)_t|\mathbf{X}_0^{Co}, t)||^2 \right] \right] + \frac{D}{2} \log \left( 2\pi e \sigma_K^2 \right)$$

$$(12)$$

Where $D$ is the dimension of the latent space. Thus, the final training objective can be expressed as:

$$\mathcal{L}(\theta) = \mathbb{E}_{\mathbf{Z}_0 \sim q(\mathbf{Z}_0|\bar{\mathbf{X}}_0^{Or}, \mathbf{X}_0^{Co})} \left[ \left\| (\hat{\mathbf{X}} - \mathbf{X}_0) \odot \mathbf{M}^{Or} \right\|^2 - \log q(\mathbf{Z}_0|\bar{\mathbf{X}}_0^{Or}, \mathbf{X}_0^{Co}) + \sum_{k=1}^{K} \log \left| \det \frac{\partial f_k}{\partial \mathbf{Z}_{k-1}} \right| \right]$$

$$+ \mathbb{E}_{t \sim \mathcal{U}[0,1]} \left[ \mathbb{E}_{q(\mathbf{Z}_0|\bar{\mathbf{X}}_0^{Or}, \mathbf{X}_0^{Co}), \boldsymbol{\epsilon} \sim \mathcal{N}(0, \mathbf{I})} \left[ ||\boldsymbol{\epsilon} - \boldsymbol{\epsilon}_\theta((\mathbf{Z}_K)_t|\mathbf{X}_0^{Co}, t)||^2 \right] \right] + \frac{D}{2} \log \left( 2\pi e \sigma_K^2 \right) \quad (13)$$

$$+ \mathbb{E}_{\boldsymbol{\epsilon} \sim \mathcal{N}(0, \mathbf{I}), t} \left\| \left( \boldsymbol{\epsilon} - \boldsymbol{\epsilon}_\theta(\mathbf{X}_t^{Ta}, t|\mathbf{X}_0^{Co}, \hat{\mathbf{X}}_0^{Or}) \right) \odot (1 - \mathbf{M}^{Ta}) \right\|^2$$

where $\hat{\mathbf{X}}$ denotes the reconstructed output of the VAE decoder. Further details on latent diffusion guidance and the continuous-time sampling procedure are provided in Appendix A.7.

## 5.3 Cross attention between encoder latent variable and reconstructed latent variable

Unlike latent diffusion models such as Stable Diffusion, which focus primarily on generation tasks and use projection to reduce computational complexity of high-dimensional observation space, our approach focuses on exploring the latent structure of the dataset and data reconstruction. In order to balance the generation and reconstruction capability of latent diffusion, we introduce a cross-attention layer Vaswani et al. (2017) to the original latent variable $\mathbf{Z}_k$ and the reconstructed

latent variable $\hat{\mathbf{Z}}_K$ of continuous latent diffusion. As a result, the output of the VAE decoder $\hat{\mathbf{X}} = \mathbf{VAE}_{dec}(\text{Attention}(\mathbf{Z}_K, \hat{\mathbf{Z}}_K))$ can be obtianed by:

$$\hat{\mathbf{X}} = \mathbf{VAE}_{dec}\left[\text{softmax}\left(\frac{QK^T}{\sqrt{D}}\right) \cdot V\right] \tag{14}$$

Where $Q = W_Q \cdot \mathbf{Z}_K, \quad K = W_K \cdot \hat{\mathbf{Z}}_K, \quad V = W_V \cdot \hat{\mathbf{Z}}_K.$ and $W_Q, W_K, W_V \in \mathbb{R}^{D \times D}$ are the learnable matrices. The overview of the algorithm is shown in Fig. 2. and the corresponding algorithms are listed in Appendix A.8.

## 6 EXPERIMENTS

**Datasets and evaluation setting:** We evaluate our method on benchmark datasets from diverse domains with varying temporal dynamics and missingness; detailed descriptions are provided in Appendix A.9. Following the out-of-sample protocol Cini et al. (2021), datasets are split into disjoint training, validation and test sequences. For P2012, MIMIC-IV, and the Synthetic dataset, we additionally mask 50% of observed points, as in CSDI Tashiro et al. (2021). For ETT, we adopt the GRIN Cini et al. (2021) block-missing strategy with a more challenging setting: random masking of 10% plus block masking of 6–24 steps with 1.5% probability. The datasets capture complementary challenges for time series imputation: P2012 Silva et al. (2012) and MIMIC-IV v3.1 Johnson et al. (2024) are large-scale clinical datasets with high natural missingness (80.52% and 49.09%), ETT Zhou et al. (2021) is fully observed but subjected to simulated structured missingness, and the Synthetic dataset Fang et al. (2024) provides controlled multiscale correlations. For ETT and Synthetic, simulated missing values are excluded from training, ensuring equal treatment of original and natural missingness and enabling analysis of ground-truth versus model-implied score functions.

**Experimental results:** As shown in Tab. 1, HSGM achieves the best imputation performance compared to all baselines. Traditional models fail to capture the nonlinear dependencies inherent in time series, while matrix completion methods struggle to identify reliable low-rank structures under severe missingness. Discriminative deep learning models such as RNNs and GNNs rely primarily on temporal or spatial neighbors for representation learning, which reduces their robustness under irregular sampling and high missing rates, as frequently encountered in healthcare data. In contrast, generative models aim to capture the underlying data distribution rather than depending solely on local neighbor information, making them more flexible for imputing realistic missing values and effectively leveraging labeled data in complex scenarios. Furthermore, methods that ignore the original missingness and directly apply observation diffusion layers, such as CSDI, exhibit limited generative capacity and induce substantial bias in the score-based function. This ultimately degrades both imputation accuracy and uncertainty estimation, whereas HSGM can flexibly handle the original missing data.

Table 1: Results of different methods across datasets

| Model | P2012@50% | | | MIMIC-IV@50% | | | ETT@Block missing | | | Synthetic dataset@50% | | |
|---|---|---|---|---|---|---|---|---|---|---|---|---|
| | MAE | RMSE | CRPS | MAE | RMSE | CRPS | MAE | RMSE | CRPS | MAE | RMSE | CRPS |
| *Traditional iterative* | | | | | | | | | | | | |
| Mean | 0.703±0.000 | 1.016±0.000 | — | 0.138±0.000 | 0.381±0.000 | — | 0.733±0.000 | 1.136±0.000 | — | 0.382±0.000 | 0.435±0.000 | — |
| KNN | 4.398±0.000 | 7.803±0.000 | — | 1.641±0.000 | 2.442±0.000 | — | 0.949±0.000 | 1.260±0.000 | — | 0.951±0.000 | 1.076±0.000 | — |
| MICE | 0.698±0.000 | 1.046±0.000 | — | 0.140±0.000 | 0.380±0.000 | — | 0.494±0.000 | 0.807±0.000 | — | 0.404±0.000 | 0.520±0.000 | — |
| *Matrix Completion* | | | | | | | | | | | | |
| MF | 1.673±0.000 | 3.899±0.000 | — | 0.230±0.000 | 0.451±0.000 | — | 0.527±0.000 | 0.725±0.000 | — | 0.173±0.000 | 0.207±0.000 | — |
| M²DMTF (ICLR 2021) | 0.700±0.001 | 1.095±0.001 | — | 0.363±0.000 | 0.980±0.001 | — | 0.544±0.001 | 0.881±0.001 | — | 0.448±0.001 | 0.497±0.001 | — |
| *Non-GNN models* | | | | | | | | | | | | |
| Transformer (NeurIPS 2017) | 0.297±0.002 | 0.675±0.027 | — | 0.058±0.001 | 0.182±0.002 | — | 0.532±0.003 | 0.933±0.004 | — | 0.113±0.009 | 0.155±0.012 | — |
| BRITS (NeurIPS 2017) | 0.368±0.002 | 0.693±0.023 | — | 0.065±0.001 | 0.215±0.002 | — | 0.556±0.003 | 0.984±0.004 | — | 0.319±0.030 | 0.354±0.030 | — |
| SAIT (ESWS 2023) | 0.296±0.002 | 0.675±0.020 | — | 0.053±0.001 | 0.178±0.002 | — | 0.405±0.003 | 0.762±0.004 | — | 0.107±0.009 | 0.147±0.010 | — |
| *GNN methods* | | | | | | | | | | | | |
| MPGRU (ICLR 2018) | 0.460±0.002 | 0.832±0.023 | — | 0.071±0.002 | 0.234±0.002 | — | 0.391±0.003 | 0.831±0.004 | — | 0.394±0.003 | 0.441±0.004 | — |
| GRIN (ICLR 2022) | 0.371±0.003 | 0.737±0.021 | — | 0.056±0.002 | 0.189±0.002 | — | 0.201±0.003 | 0.460±0.004 | — | 0.231±0.003 | 0.290±0.004 | — |
| HSPGNN (CIKM 2024) | 0.321±0.003 | 0.566±0.013 | — | 0.037±0.001 | 0.122±0.003 | — | 0.206±0.010 | 0.313±0.013 | — | 0.131±0.005 | 0.180±0.004 | — |
| *Generative models* | | | | | | | | | | | | |
| CSDI (NeurIPS 2021) | 0.301±0.002 | 0.614±0.017 | 0.330±0.002 | 0.050±0.001 | 0.178±0.002 | 0.281±0.001 | 0.227±0.004 | 0.606±0.005 | 0.165±0.003 | 0.136±0.011 | 0.204±0.012 | 0.106±0.009 |
| FGTI (NeurIPS 2024) | 0.686±0.002 | 1.708±0.012 | 0.106±0.002 | 0.055±0.001 | 0.192±0.002 | **0.063±0.001** | 0.225±0.004 | 0.418±0.005 | 0.191±0.003 | 0.143±0.015 | 0.188±0.012 | 0.164±0.008 |
| BayOTIDE (ICML 2024) | 0.548±0.002 | 0.834±0.010 | 0.497±0.002 | 0.064±0.002 | 0.147±0.003 | 0.510±0.003 | 0.332±0.002 | 0.516±0.005 | 0.495±0.003 | 0.147±0.010 | 0.181±0.002 | 0.745±0.008 |
| LSSDM (ICASSP 2025) | 0.262±0.002 | 0.598±0.015 | 0.315±0.002 | 0.042±0.001 | 0.126±0.002 | 0.251±0.002 | 0.221±0.004 | 0.585±0.005 | 0.164±0.002 | 0.112±0.018 | 0.157±0.012 | 0.086±0.009 |
| DiffPuter (ICLR 2025) | 0.496±0.004 | 0.781±0.013 | **0.067±0.003** | 0.056±0.004 | 0.160±0.003 | 0.086±0.003 | 0.605±0.005 | 1.073±0.010 | 0.086±0.003 | 0.133±0.014 | 0.199±0.010 | 0.101±0.013 |
| **HSGM (Ours)** | **0.241±0.003** | **0.538±0.015** | 0.273±0.002 | **0.032±0.002** | **0.109±0.003** | 0.205±0.002 | **0.180±0.002** | **0.289±0.004** | **0.030±0.002** | **0.104±0.010** | **0.146±0.011** | **0.077±0.008** |

[1]For GNN methods, Pearson correlation is applied.

Meanwhile, HSGM achieves comparable Continuous Ranked Probability Score (CRPS) Matheson & Winkler (1976) among the generative baselines, indicating that it not only improves imputation accuracy but also captures realistic data distributions by leveraging both observation and latent diffusion layers. Compared to CSDI, we visualize imputation results on four datasets in Fig.3. These examples demonstrate that HSGM produces more accurate imputations with tighter uncertainty estimates while maintaining consistency with the observed data, highlighting the beneficial effect of latent diffusion on the observation diffusion process. Although CRPS is slightly higher on a few

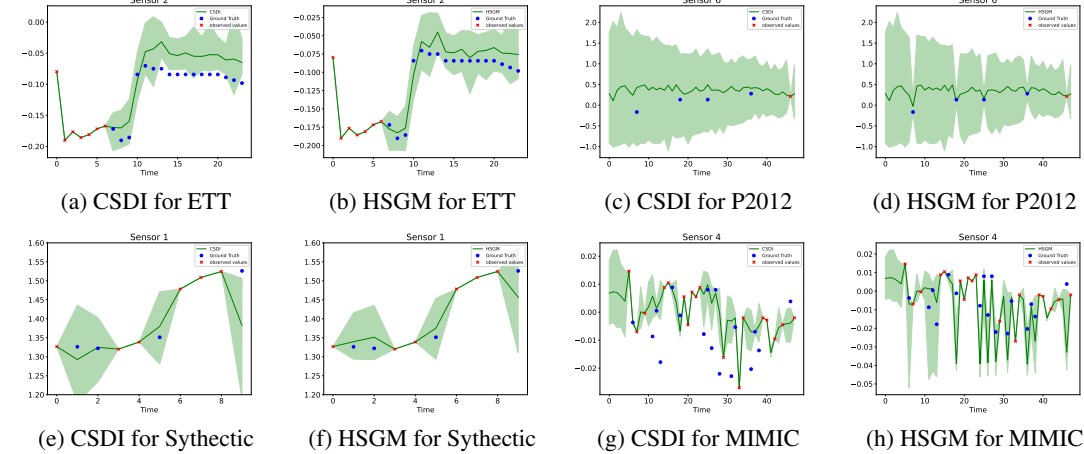

(a) CSDI for ETT     (b) HSGM for ETT     (c) CSDI for P2012     (d) HSGM for P2012

(e) CSDI for Sythetic     (f) HSGM for Sythetic     (g) CSDI for MIMIC     (h) HSGM for MIMIC

Figure 3: Probabilistic time series imputation examples across different datasets. Observed values are indicated by red crosses, and ground-truth imputation targets by blue circles. The median imputed values are shown as lines, with the 5% and 95% quantiles represented as shaded areas.

datasets, this mainly reflects HSGM's focus on accurate point estimation, which favors narrower predictive distributions. An ablation study is provided in Appendix A.14.

**Bias of score-based function:** In the Synthetic and ETT dataset settings, simulated missing values are treated as original missing data and are excluded from the training stage. Accordingly, the objective in Eq. 9 can be rewritten as

$$
\log \frac{p(\mathbf{X}_0^{Ta}|\mathbf{X}_0^{Co})}{p(\mathbf{X}_0^{Ta}|\mathbf{X}_0^{Co}, \hat{\mathbf{X}}_0^{Ta})} \approx \log \int p(\hat{\mathbf{X}}_0^{Ta}|\mathbf{X}_0^{Co}, \mathbf{Z}_0)\, p(\mathbf{Z}_0|\mathbf{X}_0^{Co})\, d\mathbf{Z}_0,
$$
$$
\leqslant \mathbb{E}_q\Big[ -\log p(\hat{\mathbf{X}}_0^{Or}|\mathbf{X}_0^{Co}, \mathbf{Z}_K) + \log q(\mathbf{Z}_0|\bar{\mathbf{X}}_0^{Or}, \mathbf{X}_0^{Co}) - \sum_{k=1}^{K} \log \left| \det \frac{\partial f_k}{\partial \mathbf{Z}_{k-1}} \right| - \log p(\mathbf{Z}_K|\mathbf{X}_0^{Co}) \Big].
$$

(15)

Interestingly, by moving the imputed distribution term to the left-hand side, one obtains the likelihood ratio between the ground-truth and imputed distributions, which can be estimated via the latent generative term. This observation suggests that future improvements in latent generative imputation should focus on optimizing the latent-space objective to maximize this probabilistic ratio, potentially yielding more principled and effective imputation strategies. The latent variable $\mathbf{Z}_K$ fundamentally governs the information gain in Eq. 15 through its learned representation. Consequently, higher ratios correspond to more confident imputations, whereas lower ratios indicate greater uncertainty in the imputation process. In this study, $\log p(\mathbf{X}_0^{Ta}|\mathbf{X}_0^{Co})$ and $p(\mathbf{X}_0^{Ta}|\mathbf{X}_0^{Co}, \hat{\mathbf{X}}_0^{Ta})$ are sampled using the learned score-based function as in Eq. 5. To investigate the effect of missing data on the score-based function, we train CSDI on the Ground truth data, with missing values replaced by zeros, and HSGM on the Synthetic training dataset while evaluating them on the same test dataset. Fig. 4a and 4b present the MAE of bias and the accumulated bias of Eq. 5 along the reverse process. Initially, all learned score-based functions yield nearly identical values; however, biases grow progressively over the reverse steps, consistent with the theoretical analysis in Fig. 1. Correcting such bias remains challenging: as reverse time increases, accumulated bias amplifies, potentially leading to divergence from the true distribution. Notably, HSGM effectively mitigates this bias, achieving superior performance compared to CSDI, as also reflected in Tab. 1. To further illustrate this effect, we reshape and visualize the heat maps of the learned score-based functions at different reverse time steps in Fig. 4 (c-k). These visualizations demonstrate that HSGM consistently corrects bias in the score-based function. Similar visualizations for the ETT dataset are provided in Appendix A.13.

**Generation vs Reconstruction:** High-fidelity and diverse imputations for missing values often rely on unconditional score-based functions with latent diffusion, while reconstruction of observed (non-missing) values can benefit from conditional score-based functions and a VAE architecture. For datasets containing both missing and observed values, balancing generative and reconstructive capabilities is crucial. In this work, we address this challenge by combining the original and denoised latent variables through cross-attention mechanisms, together with continuous latent diffusion and ODE-based sampling in the latent space. To evaluate the effectiveness of this approach, we compare it against the following baselines: (1) Conditional latent probability flow (PF) ODE, (2) Unconditional latent PF ODE. (3) VAE architecture with norm flow. (4) Conditional latent PF ODE with cross attention (Ours). The results are summarized in Tab. 2. As shown in Tab. 2, our model achieves

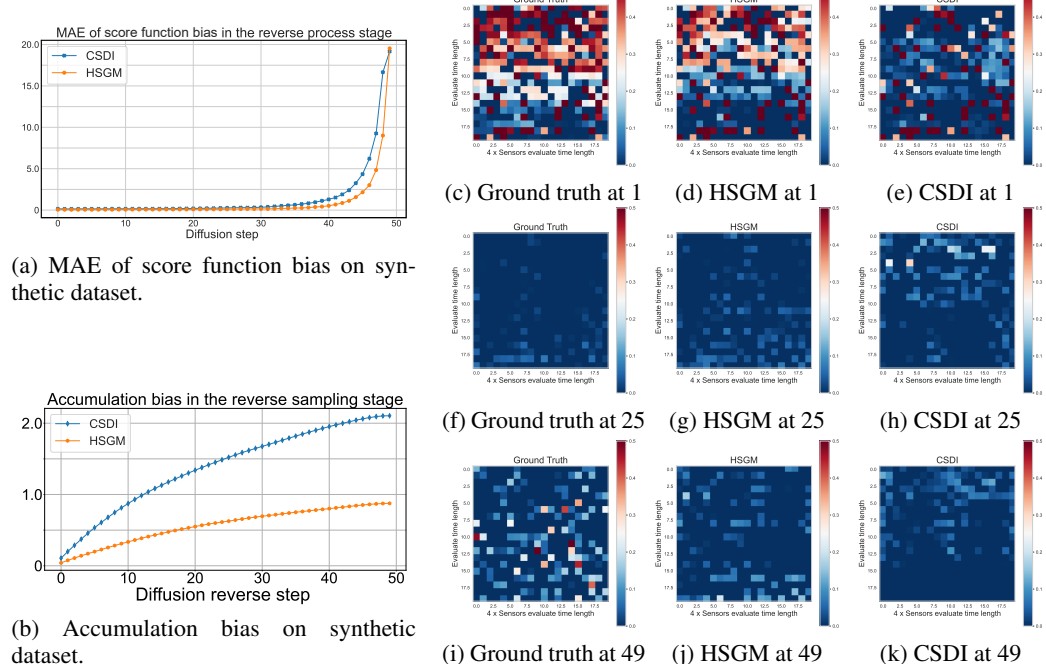

(a) MAE of score function bias on synthetic dataset.

(b) Accumulation bias on synthetic dataset.

(c) Ground truth at 1    (d) HSGM at 1    (e) CSDI at 1

(f) Ground truth at 25    (g) HSGM at 25    (h) CSDI at 25

(i) Ground truth at 49    (j) HSGM at 49    (k) CSDI at 49

Figure 4: Synthetic dataset evaluation: (a) MAE of bias; (b) accumulated bias over the reverse process; (c–k) heat maps of learned score-based functions at selected reverse time steps for ground truth, HSGM, and CSDI.

Table 2: Generation and reconstruction performance of generative models on benchmark datasets

| Datasets | Models | Generation | | Reconstruction | |
|---|---|---|---|---|---|
| | | MAE | RMSE | MAE | RMSE |
| P2012@50% | Conditional latent PF ODE | 1.703±0.003 | 1.979±0.015 | 2.040±0.003 | 2.316±0.015 |
| | Unconditional latent PF ODE | 1.829±0.004 | 2.103±0.016 | 2.207±0.004 | 2.499±0.016 |
| | VAE-norm | 0.374±0.003 | 0.610±0.015 | 0.326±0.003 | 0.536±0.015 |
| | **PF ODE with cross attention (Ours)** | **0.329±0.003** | **0.569±0.015** | **0.227±0.003** | **0.449±0.015** |
| MIMIC IV @50% | Conditional latent PF ODE | 0.167±0.003 | 0.325±0.004 | 0.224±0.003 | 0.413±0.004 |
| | Unconditional latent PF ODE | 0.194±0.003 | 0.362±0.004 | 0.257±0.003 | 0.461±0.004 |
| | VAE-norm | 0.045±0.002 | 0.129±0.003 | 0.042±0.002 | 0.121±0.003 |
| | **PF ODE with cross attention (Ours)** | **0.041±0.002** | **0.122±0.003** | **0.032±0.002** | **0.081±0.003** |
| ETT@Block missing | Conditional latent PF ODE | 0.941±0.003 | 1.227±0.005 | 0.963±0.004 | 1.258±0.005 |
| | Unconditional latent PF ODE | 0.931±0.003 | 1.218±0.005 | 0.973±0.004 | 1.269±0.005 |
| | VAE-norm | 0.190±0.002 | 0.294±0.004 | 0.123±0.003 | 0.181±0.004 |
| | **PF ODE with cross attention (Ours)** | **0.180±0.002** | **0.289±0.004** | **0.103±0.003** | **0.148±0.004** |
| Synthetic@50% | Conditional latent PF ODE | 0.491±0.012 | 0.560±0.017 | 0.463±0.012 | 0.532±0.018 |
| | Unconditional latent PF ODE | 0.532±0.011 | 0.556±0.016 | 0.460±0.013 | 0.529±0.017 |
| | VAE-norm | 0.190±0.010 | 0.234±0.012 | 0.181±0.010 | 0.223±0.012 |
| | **PF ODE with cross attention (Ours)** | **0.173±0.010** | **0.215±0.014** | **0.153±0.010** | **0.193±0.015** |

the best performance in both generation and reconstruction. A key advantage of the PF ODE with cross-attention is its ability to balance these two objectives. By leveraging cross-attention over latent space, the model selectively emphasizes informative patterns, enabling realistic sequence generation while maintaining fidelity to observed data. In contrast, VAE-based models typically prioritize reconstruction at the expense of generative diversity, whereas latent PF ODE models may generate plausible sequences but struggle to reconstruct observed values accurately.

# 7 CONCLUSION

We theoretically analyze bias in the score function induced by missing data within the DDPM framework, showing that ignoring missing patterns—especially under high missing rates or strong inter-variable correlations—can significantly impair the learned score function. To address this, we propose HSGM, which bridges observation and latent diffusion via the chain rule of probability and unsupervised VAE projection. Flexible latent distributions are modeled through normalizing flows, while cross-attention between original and denoised latent variables balances generative and reconstructive capabilities. Our model effectively mitigates score function bias, yielding more accurate imputations with reduced uncertainty. Experiments verify that HSGM consistently surpasses prior methods, demonstrating its effectiveness.

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

# A  APPENDIX

**Use of large language models statement**  We use the LLM to polish the writing. All other parts, including experimental results, analyses were written by the authors and carefully verified for accuracy before and after any LLM-assisted editing.

## A.1  VISUALIZATION OF AVAILABLE DATA MATRIX

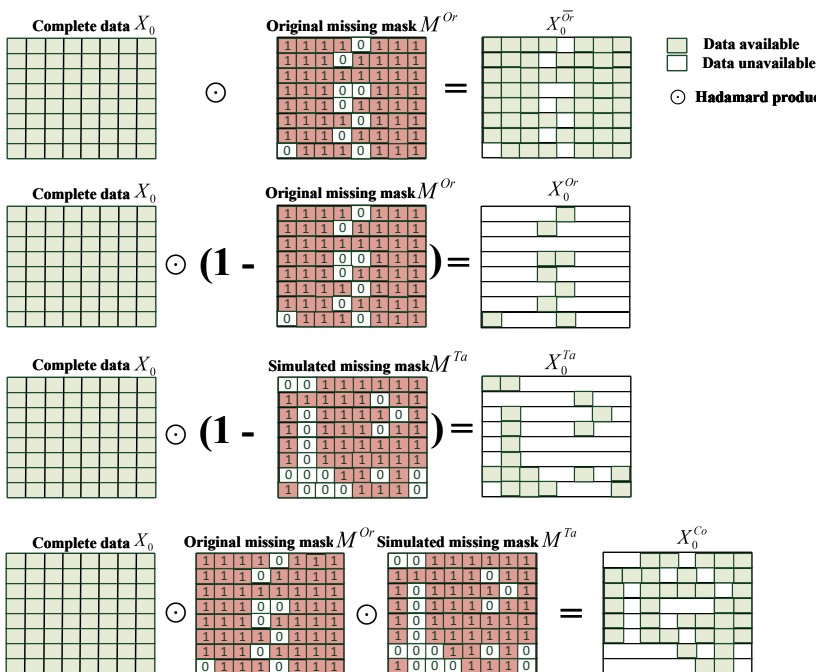

Figure 5: Illustration of how the available data matrix is created.

## A.2  PROOF OF PROPOSITION 4.1

*Proof.* According to the assumption, $\forall x_{0(ij)}$, $\forall x_{0(kl)} \in \mathbf{X}_0$, $x_{0(ij)} \perp\!\!\!\perp x_{0(kl)}$. As a result, we will obtain:

$$p(\mathbf{X}_0) = \prod_{i=0}^{N}\prod_{j=0}^{F} p(x_{0(ij)}) \tag{16}$$

where $x_{0(ij)}$ is the signal of $i$-th sensor in the $j$-th time step of $\mathbf{X}_0$. With this assumption, the true score-based function can be obtained by a variational Markov chain as:

$$
\begin{aligned}
s_{ij}(x_{t(ij)}, t) &= \nabla_{x_{t(ij)}} \log \int q(x_{t(ij)}|x_{0(ij)})p(x_{0(ij)})dx_{0(ij)} \\
&= \frac{\nabla_{x_{t(ij)}} \int q(x_{t(ij)}|x_{0(ij)})p(x_{0(ij)})dx_{0(ij)}}{\int q(x_{t(ij)}|x_{0(ij)})p(x_{0(ij)})dx_{0(ij)}} \\
&= \frac{\int p(x_{0(ij)})\nabla_{x_{t(ij)}}\mathcal{N}(x_{t(ij)} : \sqrt{\bar{\alpha}_t}x_{0(ij)}, (1-\bar{\alpha}_t))dx_{0(ij)}}{\int \mathcal{N}(x_{t(ij)} : \sqrt{\bar{\alpha}_t}x_{0(ij)}, (1-\bar{\alpha}_t))p(x_{0(ij)})dx_{0(ij)}} \\
&= \frac{\int p(x_{0(ij)})(-\frac{x_{t(ij)}-x_{0(ij)}}{1-\bar{\alpha}_t})\mathcal{N}(x_{t(ij)} : \sqrt{\bar{\alpha}_t}x_{0(ij)}, (1-\bar{\alpha}_t))dx_{0(ij)}}{\int \mathcal{N}(x_{t(ij)} : \sqrt{\bar{\alpha}_t}x_{0(ij)}, (1-\bar{\alpha}_t))p(x_{0(ij)})dx_{0(ij)}} \\
&= -\frac{x_{t(ij)}-x_{0(ij)}}{1-\bar{\alpha}_t} = -\frac{\epsilon}{\sqrt{1-\bar{\alpha}_t}}
\end{aligned}
\tag{17}
$$

The learning label is the same as a non-missing DDPM score-based function, but the input of the score-based function is changed. Then, we apply Taylor expansion to $s_{ij}(x_{t(ij)}, t)$ at the point $x_{t(ij)}^{\overline{Or}}$ as:

$$s_{ij}(x_{t(ij)}, t) = s_{ij}(x_{t(ij)}^{\overline{Or}}, t) + \frac{s_{ij}'(x_{t(ij)}^{\overline{Or}}, t)}{1!}(x_{t(ij)} - x_{t(ij)}^{\overline{Or}}) + \frac{s_{ij}''(x_{t(ij)}^{\overline{Or}}, t)}{2!}(x_{t(ij)} - x_{t(ij)}^{\overline{Or}})^2$$
$$+ ... + \frac{s_{ij}^{(n)}(x_{t(ij)}^{\overline{Or}}, t)}{n!}(x_{t(ij)} - x_{t(ij)}^{\overline{Or}})^n + R_n(x_{t(ij)}, t) \tag{18}$$

$$\because s_{ij}'(x_{t(ij)}, t) = \nabla_{x_{t(ij)}} \nabla_{x_{t(ij)}} \log \int q(x_{t(ij)}|x_{0(ij)})p(x_{0(ij)})dx_{0(ij)}$$

$$= \nabla_{x_{t(ij)}} \frac{\nabla_{x_{t(ij)}} \int q(x_{t(ij)}|x_{0(ij)})p(x_{0(ij)})dx_{0(ij)}}{\int q(x_{t(ij)}|x_{0(ij)})p(x_{0(ij)})dx_{0(ij)}}$$

$$= \nabla_{x_{t(ij)}} \frac{\int p(x_{0(ij)})\nabla_{x_{t(ij)}} \mathcal{N}(x_{t(ij)} : \sqrt{\bar{\alpha}_t}x_{0(ij)}, (1-\bar{\alpha}_t))dx_{0(ij)}}{\int \mathcal{N}(x_{t(ij)} : \sqrt{\bar{\alpha}_t}x_{0(ij)}, (1-\bar{\alpha}_t))p(x_{0(ij)})dx_{0(ij)}}$$

$$= \nabla_{x_{t(ij)}} \frac{\int p(x_{0(ij)})(-\frac{x_{t(ij)} - \sqrt{\bar{\alpha}_t}x_{0(ij)}}{1-\bar{\alpha}_t})\mathcal{N}(x_{t(ij)} : \sqrt{\bar{\alpha}_t}x_{0(ij)}, (1-\bar{\alpha}_t))dx_{0(ij)}}{\int \mathcal{N}(x_{t(ij)} : \sqrt{\bar{\alpha}_t}x_{0(ij)}, (1-\bar{\alpha}_t))p(x_{0(ij)})dx_{0(ij)}}$$

$$= \nabla_{x_{t(ij)}}[-\frac{x_{t(ij)} - \sqrt{\bar{\alpha}_t}x_{0(ij)}}{1-\bar{\alpha}_t}] = -\frac{1}{1-\bar{\alpha}_t}$$

$$s_{ij}''(x_{t(ij)}, t) = \nabla_{x_{t(ij)}}[-\frac{1}{1-\bar{\alpha}_t}] = 0$$

$$....$$

$$s_{ij}^n(x_{t(ij)}, t) = 0$$

$$\therefore s_{ij}(x_{t(ij)}, t) = s_{ij}(x_{t(ij)}^{\overline{Or}}, t) + \frac{s_{ij}'(x_{t(ij)}^{\overline{Or}}, t)}{1!}(x_{t(ij)} - x_{t(ij)}^{\overline{Or}}) \tag{19}$$

Thus, we can obtain:

$$\Rightarrow \mathbf{s}_{bias(ij)} = s_{ij}(x_{t(ij)}, t) - s_{ij}(x_{t(ij)}^{\overline{Or}}, t) = s_{ij}'(x_{t(ij)}^{\overline{Or}}, t)\sqrt{\bar{\alpha}_t}x_{0(ij)} = -\frac{\sqrt{\bar{\alpha}_t}x_{0(ij)}}{1-\bar{\alpha}_t} \tag{20}$$

$$\square$$

### A.3 PROOF OF COROLLARY 4.1

*Proof.* Considering the original missing rate as $p^{Or} = \frac{\sum_{i=0}^{N}\sum_{j=0}^{F}(1-\mathbf{M}_{ij}^{Or})}{NF}$, We analyze the cumulative error in DDPM sampling process, focusing on the bias introduced by finite-difference approximations in score estimation. From Eq. 5, each step's error propagates through subsequent steps with amplification factor $\frac{\beta_t}{\sqrt{1-\beta_t}}$, which can be transformed as $\frac{1-\alpha_t}{\sqrt{\alpha_t}}$. The expectation of the accumulated bias can be obtained as:

$$\mathbb{E}\left[\sum_{t=1}^{T} \mathbf{s}_{bias}\right] = \mathbb{E}\left[\sum_{t=1}^{T} \left\|\mathbf{s}(\mathbf{X}_t) - \mathbf{s}(\mathbf{X}_t^{\overline{Or}}, t)\right\|^2\right]$$

$$= \mathbb{E}\left[\sum_{t=1}^{T} \left\|\frac{1-\alpha_t}{\sqrt{\alpha_t}}\mathbf{s}_{bias} \odot (1-\mathbf{M}^{Or})\right\|^2\right]$$

$$= p^{Or}[\sum_{t=1}^{T} \sum_{i,j \in Or} \left[-\frac{1-\alpha_t}{\sqrt{\alpha_t}}\frac{\sqrt{\bar{\alpha}_t}}{1-\bar{\alpha}_t}x_{0(ij)}\right]$$

$$= p^{Or} \sum_{t=1}^{T} \sum_{i,j \in Or} \left[-\sqrt{\bar{\alpha}_{t-1}}\frac{1-\alpha_t}{1-\bar{\alpha}_t}x_{0(ij)}\right] \tag{21}$$

$$\square$$

Where Or means the original missing data.

### A.4 PROOF OF PROPOSITION 4.2

*Proof.* The real situation will be more complex since most of the time series data possess temporal correlation or even spatio-temporal relationships. In this case, the indpendent assumption of Proposition 4.1 will not be valid due to $x_{0(ij)} \not\perp x_{0(kl)}$. Therefore, the original missing bias can provably cause the bias of a non-original missing entity. To analyze this effect, we analyze two correlated points $\mathbf{X}_0 = \begin{bmatrix} x_{0(ij)} \\ x_{0(kl)} \end{bmatrix}$ with a non-independent Gaussian distribution in the DDPM setting. The probability density function is:

$$p(\mathbf{X}_0) = \frac{1}{2\pi\sqrt{|\boldsymbol{\Sigma}_0|}} \exp\left(-\frac{1}{2}(\mathbf{X}_0 - \boldsymbol{\mu})^\top \boldsymbol{\Sigma}_0^{-1}(\mathbf{X}_0 - \boldsymbol{\mu})\right), \tag{22}$$

where mean vector $\boldsymbol{\mu} = \begin{bmatrix} \mu_{ij} \\ \mu_{kl} \end{bmatrix}$ and $\boldsymbol{\Sigma}_0 = \begin{bmatrix} \sigma_{ij}^2 & \rho\sigma_{ij}\sigma_{kl} \\ \rho\sigma_{ij}\sigma_{kl} & \sigma_{kl}^2 \end{bmatrix}$ with correlation coefficient $\rho \in [-1, 1]$ . Then, in the forward process, $p(\mathbf{X}_t|\mathbf{X}_0) \sim \mathcal{N}(\mathbf{X}_t; \sqrt{\bar{\alpha}_t}\mathbf{X}_0, (1 - \bar{\alpha}_t)\mathbf{I})$ and $p(\mathbf{X}_0) = \mathcal{N}(\mathbf{X}_0; \boldsymbol{\mu}, \boldsymbol{\Sigma}_0)$, let $\boldsymbol{\Sigma}_t = \bar{\alpha}_t\boldsymbol{\Sigma}_0 + (1 - \bar{\alpha}_t)\mathbf{I}$, we can obtain $p(\mathbf{X}_t) \sim \mathcal{N}(\mathbf{X}_t; \sqrt{\bar{\alpha}_t}\boldsymbol{\mu}, \boldsymbol{\Sigma}_t)$. Thus, the score-based function can be calculated as:

$$\mathbf{s}(\mathbf{X}_t, t) = \nabla_{\mathbf{X}_t} \log p(\mathbf{X}_t) = \nabla_{\mathbf{X}_t} \log \mathcal{N}(\mathbf{X}_t; \sqrt{\bar{\alpha}_t}\boldsymbol{\mu}, \boldsymbol{\Sigma}_t)$$

$$= \nabla_{\mathbf{X}_t}\left[-\frac{1}{2}(\mathbf{X}_t - \sqrt{\bar{\alpha}_t}\boldsymbol{\mu})^T \boldsymbol{\Sigma}_t^{-1}(\mathbf{X}_t - \sqrt{\bar{\alpha}_t}\boldsymbol{\mu}) + \text{constant}\right]$$

$$= -\frac{1}{2}(\boldsymbol{\Sigma}_t^{-1} + (\boldsymbol{\Sigma}_t^{-1})^T)(\mathbf{X}_t - \sqrt{\bar{\alpha}_t}\boldsymbol{\mu})$$

$$= -\boldsymbol{\Sigma}_t^{-1}(\mathbf{X}_t - \sqrt{\bar{\alpha}_t}\boldsymbol{\mu})$$

$$= \frac{1}{D_t}\begin{bmatrix} \bar{\alpha}_t\sigma_{kl}^2 + (1 - \bar{\alpha}_t) & -\bar{\alpha}_t\rho\sigma_{ij}\sigma_{kl} \\ -\bar{\alpha}_t\rho\sigma_{ij}\sigma_{kl} & \bar{\alpha}_t\sigma_{ij}^2 + (1 - \bar{\alpha}_t) \end{bmatrix}\left(\begin{bmatrix} x_{t(ij)} \\ x_{t(kl)} \end{bmatrix} - \sqrt{\bar{\alpha}_t}\begin{bmatrix} \mu_{ij} \\ \mu_{kl} \end{bmatrix}\right)$$

$$= \frac{1}{D_t}\begin{bmatrix} \bar{\alpha}_t\sigma_{kl}^2 + (1 - \bar{\alpha}_t) & -\bar{\alpha}_t\rho\sigma_{ij}\sigma_{kl} \\ -\bar{\alpha}_t\rho\sigma_{ij}\sigma_{kl} & \bar{\alpha}_t\sigma_{ij}^2 + (1 - \bar{\alpha}_t) \end{bmatrix}\begin{bmatrix} x_{t(ij)} - \sqrt{\bar{\alpha}_t}\mu_{ij} \\ x_{t(kl)} - \sqrt{\bar{\alpha}_t}\mu_{kl} \end{bmatrix}$$

$$= \frac{1}{D_t}\begin{bmatrix} (1 + \bar{\alpha}_t(\sigma_{kl}^2 - 1))x_{t(ij)} - \bar{\alpha}_t\rho\sigma_{ij}\sigma_{kl}x_{t(kl)} - (1 + \bar{\alpha}_t(\sigma_{kl}^2 - 1))\sqrt{\bar{\alpha}_t}\,\mu_{ij} + \bar{\alpha}_t\rho\sigma_{ij}\sigma_{kl}\sqrt{\bar{\alpha}_t}\,\mu_{kl} \\ -\bar{\alpha}_t\rho\sigma_{ij}\sigma_{kl}x_{t(ij)} + (1 + \bar{\alpha}_t(\sigma_{ij}^2 - 1))x_{t(kl)} + \bar{\alpha}_t\rho\sigma_{ij}\sigma_{kl}\sqrt{\bar{\alpha}_t}\,\mu_{ij} - (1 + \bar{\alpha}_t(\sigma_{ij}^2 - 1))\sqrt{\bar{\alpha}_t}\,\mu_{kl} \end{bmatrix}$$

$$\tag{23}$$

Where $D_t = \text{Determinant}(\boldsymbol{\Sigma}_t) = (\bar{\alpha}_t\sigma_{ij}^2 + (1 - \bar{\alpha}_t))(\bar{\alpha}_t\sigma_{kl}^2 + (1 - \bar{\alpha}_t)) - (\bar{\alpha}_t\rho\sigma_{ij}\sigma_{kl})^2$. From Eq. 23, if the two points do not correlate with each other, i.e., $\rho = 0$, and the score-based function can be transformed as:

$$\mathbf{s}(\mathbf{X}_t, t) = \frac{1}{(\bar{\alpha}_t\sigma_{ij}^2 + (1 - \bar{\alpha}_t))(\bar{\alpha}_t\sigma_{kl}^2 + (1 - \bar{\alpha}_t))}\begin{bmatrix} (1 + \bar{\alpha}_t(\sigma_{kl}^2 - 1))x_{t(ij)} - (1 + \bar{\alpha}_t(\sigma_{kl}^2 - 1))\sqrt{\bar{\alpha}_t}\,\mu_{ij} \\ (1 + \bar{\alpha}_t(\sigma_{ij}^2 - 1))x_{t(kl)} - (1 + \bar{\alpha}_t(\sigma_{ij}^2 - 1))\sqrt{\bar{\alpha}_t}\,\mu_{kl} \end{bmatrix}$$

$$= \begin{bmatrix} \frac{x_{t(ij)} - \sqrt{\bar{\alpha}_t}\mu_{ij}}{\bar{\alpha}_t\sigma_{ij}^2 + (1 - \bar{\alpha}_t)} \\ \frac{x_{t(kl)} - \sqrt{\bar{\alpha}_t}\mu_{kl}}{\bar{\alpha}_t\sigma_{kl}^2 + (1 - \bar{\alpha}_t)} \end{bmatrix}$$

$$\tag{24}$$

and there is no effect on one point, even the other point suffers the original missing and is replaced with 0, which has the same result and conclusion as Proposition 4.1. We assume that $x_{0(kl)}$ is the original missing point while $x_{0(ij)}$ is the observed point. Thus, $\rho^{\overline{Or}} = 0$, $\mu_{kl}^{\overline{Or}} = 0$ and $\sigma_{kl}^{\overline{Or}} = 0$, then, $s_{ij}(x_t^{\overline{Or}}, t) = \frac{x_{t(ij)} - \sqrt{\bar{\alpha}_t}\mu_{ij}}{\bar{\alpha}_t\sigma_{ij}^2 + (1 - \bar{\alpha}_t)}$. Thus, we can obtain:

$$\mathbf{s}_{bias(ij)} = s_{ij}(x_t, t) - s_{ij}(x_t^{\overline{Or}}, t)$$

$$= \frac{1}{D_t}\left[(\bar{\alpha}_t\sigma_{kl}^2 + (1 - \bar{\alpha}_t))(x_{t(ij)} - \sqrt{\bar{\alpha}_t}\mu_{ij}) - \bar{\alpha}_t\rho\sigma_{ij}\sigma_{kl}(x_{t(kl)} - \sqrt{\bar{\alpha}_t}\mu_{kl})\right] - \frac{x_{t(ij)} - \sqrt{\bar{\alpha}_t}\mu_{ij}}{\bar{\alpha}_t\sigma_{ij}^2 + (1 - \bar{\alpha}_t)}$$

$$\tag{25}$$

From Eq. 25, $x_{t(ij)}$ still suffer the missing effect of $x_{t(kl)}$ when the correlation coefficient $\rho$ is not equal to 0, which means that even the observed data can not be reconstructed without bias if the

original missing data is correlated to the observed data. In order to know the bias effect of score-based function from the original missing data to the non-missing data, assuming all variances are unity, $\sigma_{ij}^2 = \sigma_{kl}^2 = 1$, we obtain:

$$\bar{\alpha}_t \sigma_{ij}^2 + (1 - \bar{\alpha}_t) = 1, \quad \bar{\alpha}_t \sigma_{kl}^2 + (1 - \bar{\alpha}_t) = 1, \quad \bar{\alpha}_t \rho \sigma_{ij} \sigma_{kl} = \bar{\alpha}_t \rho, \tag{26}$$

which reduces the determinant to

$$D_t = 1 - (\bar{\alpha}_t \rho)^2. \tag{27}$$

Combining these expressions and simplifying the numerator yields the final form:

$$\mathbf{s}_{bias(ij)} = \frac{(\bar{\alpha}_t \rho)^2 (x_{t(ij)} - \sqrt{\bar{\alpha}_t} \mu_{ij}) - \bar{\alpha}_t \rho (x_{t(kl)} - \sqrt{\bar{\alpha}_t} \mu_{kl})}{1 - (\bar{\alpha}_t \rho)^2}$$
$$\Longrightarrow \frac{\mathbf{s}_{bias(ij)}}{\epsilon} = \frac{(\bar{\alpha}_t \rho)^2 - \bar{\alpha}_t \rho}{1 - (\bar{\alpha}_t \rho)^2} = \frac{-\bar{\alpha}_t \rho}{1 + \bar{\alpha}_t \rho} \tag{28}$$

$\square$

The plot of Eq. 28 is shown in Fig. 1b.

### A.5 OBSERVATION SCORE-BASED OBJECTIVE FUNCTION DEDUCTION

From CSDI Tashiro et al. (2021), the objective function $\mathcal{L}_1$ can be transformed as:

$$-\log p(\mathbf{X}_0^{Ta} | \mathbf{X}_0^{Co}, \hat{\mathbf{X}}_0^{Or}) \leqslant \mathbb{E}_{q(\mathbf{X}_{1:T}^{Ta} | \mathbf{X}_0^{Co}, \hat{\mathbf{X}}_0^{Or})} \log \frac{q(\mathbf{X}_{1:T}^{Ta} | \mathbf{X}_0^{Co}, \hat{\mathbf{X}}_0^{Or})}{p_\theta(\mathbf{X}_{0:T}^{Ta} | \mathbf{X}_0^{Co}, \hat{\mathbf{X}}_0^{Or})} \tag{29}$$

Apply reparameterization as CSDI Tashiro et al. (2021), we can obtain the loss function from the observation space as:

$$\mathcal{L}_1(\theta) = \mathbb{E}_{\boldsymbol{\epsilon} \sim \mathcal{N}(0, \mathbf{I}), t} \left\| \left( \boldsymbol{\epsilon} - \boldsymbol{\epsilon}_\theta(\mathbf{X}_t^{Ta}, t | \mathbf{X}_0^{Co}, \hat{\mathbf{X}}_0^{Or}) \right) \odot (1 - \mathbf{M}^{Ta}) \right\|^2 \tag{30}$$

### A.6 LATENT SCORE-BASED OBJECTIVE FUNCTION DEDUCTION

As for the second term $\mathcal{L}_2$, we applied Jensen's inequality,

$$\begin{aligned}
\mathcal{L}_2 &= -\log \int p(\hat{\mathbf{X}}_0^{Or} | \mathbf{X}_0^{Co}, \mathbf{Z}_0) p(\mathbf{Z}_0 | \mathbf{X}_0^{Co}) d\mathbf{Z}_0 \\
&= -\log \int \frac{q(\mathbf{Z}_0 | \bar{\mathbf{X}}_0^{Or}, \mathbf{X}_0^{Co}) p(\hat{\mathbf{X}}_0^{Or} | \mathbf{X}_0^{Co}, \mathbf{Z}_0) p(\mathbf{Z}_0 | \mathbf{X}_0^{Co})}{q(\mathbf{Z}_0 | \bar{\mathbf{X}}_0^{Or}, \mathbf{X}_0^{Co})} d\mathbf{Z}_0 \\
&\leqslant \mathbb{E}_{q(\mathbf{Z}_0 | \bar{\mathbf{X}}_0^{Or}, \mathbf{X}_0^{Co})} \log \frac{q(\mathbf{Z}_0 | \bar{\mathbf{X}}_0^{Or}, \mathbf{X}_0^{Co})}{p(\hat{\mathbf{X}}_0^{Or} | \mathbf{X}_0^{Co}, \mathbf{Z}_0) p(\mathbf{Z}_0 | \mathbf{X}_0^{Co})} \\
&= -\mathbb{E}_{q(\mathbf{Z}_0 | \bar{\mathbf{X}}_0^{Ta}, \mathbf{X}_0^{Co})} \left[ \log p(\hat{\mathbf{X}}_0^{Or} | \mathbf{X}_0^{Co}, \mathbf{Z}_0) \right] + \mathbf{KL} \left( q(\mathbf{Z}_0 | \bar{\mathbf{X}}_0^{Or}, \mathbf{X}_0^{Co}) || p(\mathbf{Z}_0 | \mathbf{X}_0^{Co}) \right),
\end{aligned} \tag{31}$$

where KL denotes the Kullback–Leibler divergence, and $\bar{\mathbf{X}}^{Or}$ represents the linear interpolation values of the original missing data during the data preprocessing stage, aiming to prevent the neural network from encountering sparse input data issues. In our study, the prior distribution $p(\mathbf{Z}_0 | \mathbf{X}_0^{Co})$ and the posterior distribution $q(\mathbf{Z}_0 | \bar{\mathbf{X}}_0^{Or}, \mathbf{X}_0^{Co})$ are not as simple as the Normal distribution under complex temporal correlation and serious missing condition.

### A.6.1 NORMALIZING FLOW OBJECTIVE FUNCTION DEDUCTION

To approximate the posterior distribution $q(\mathbf{Z}_0|\bar{\mathbf{X}}_0^{Or}, \mathbf{X}_0^{Co})$, normalizing flow is applied. Given a base latent variable $\mathbf{Z}_0 \sim q(\mathbf{Z}_0|\bar{\mathbf{X}}_0^{Or}, \mathbf{X}_0^{Co})$, a normalizing flow applies a sequence of invertible, differentiable transformations Rezende & Mohamed (2015):

$$\log q(\mathbf{Z}_K|\bar{\mathbf{X}}_0^{Or}, \mathbf{X}_0^{Co}) = \log q(\mathbf{Z}_0|\bar{\mathbf{X}}_0^{Or}, \mathbf{X}_0^{Co}) - \sum_{k=1}^{K} \log \left| \det \frac{\partial f_k}{\partial \mathbf{Z}_{k-1}} \right|, \tag{32}$$

where $\det$ is determinant and $\mathbf{Z}_K = f_K \circ f_{K-1} \circ \cdots \circ f_1(\mathbf{Z}_0)$ as a shorthand for the composition $f_K(f_{K-1}(\ldots f_1(x)))$ and the normalizing flow can approximate any distribution in theory. In this study, we adopt planar flow $f$ and initialization $\mathbf{Z}_0$ as:

$$f(\mathbf{Z}) = \mathbf{Z} + \mathbf{w}_0 \tanh(\mathbf{w}_1^\top \mathbf{Z} + \mathbf{b}) \quad \mathbf{Z}_0 \sim \mathcal{N}(0, \mathbf{I}), \tag{33}$$

where $\mathbf{w}_0 \in \mathbb{R}^D, \mathbf{w}_1 \in \mathbb{R}^D$ and $\mathbf{b} \in \mathbb{R}^1$ are the learnable parameters and $D$ is the dimension of the latent space. As a result, $\mathcal{L}_2$ can be transformed to:

$$\mathcal{L}_2 = \mathbb{E}_{\mathbf{Z}_0 \sim q(\mathbf{Z}_0|\bar{\mathbf{X}}_0^{Or}, \mathbf{X}_0^{Co})} \left[ -\log p(\mathbf{X}_0^{Or}|\mathbf{X}_0^{Co}, \mathbf{Z}_K) + \log q(\mathbf{Z}_0|\bar{\mathbf{X}}_0^{Or}, \mathbf{X}_0^{Co}) - \sum_{k=1}^{K} \log \left| \det \frac{\partial f_k}{\partial \mathbf{Z}_{k-1}} \right| - \log p(\mathbf{Z}_K|\mathbf{X}_0^{Co}) \right] \tag{34}$$

## A.7 GUIDANCE AND SAMPLING STRATEGY IN LATENT DIFFUSION

### A.7.1 POSITION ENCODING

To effectively handle sequential data with missing values, we incorporate positional encoding and masked self-attention mechanisms in our model. Since the attention architecture lacks inherent positional awareness, we apply sinusoidal positional encoding to the input sequence $\mathbf{X}^{Co}$. The positional encoding vector $\mathbf{PE}$ is defined as Vaswani et al. (2017):

$$\mathbf{PE}(pos, 2i) = \sin\left(\frac{pos}{10000^{\frac{2i}{F}}}\right), \quad \mathbf{PE}(pos, 2i+1) = \cos\left(\frac{pos}{10000^{\frac{2i}{F}}}\right) \tag{35}$$

where $pos \in \{0, 1, \ldots, N_p - 1\}$ is the token index and $F_p$ is the embedding dimension while $i \in \{0, 1, \ldots, F_p/2 - 1\}$ is the channel index. The positional encodings $\mathbf{PE}$ are added to the input $\mathbf{X}_{\text{pos}} = \mathbf{X}^{Co} + \mathbf{PE}$. We then apply a multi-head self-attention mechanism with $H = 8$ heads:

$$\text{MultiHead}(Q, K, V) = \text{Concat}(h_1, \ldots, h_H)W^O, \tag{36}$$

$$h_i = \text{Attention}(QW_i^Q, KW_i^K, VW_i^V), \tag{37}$$

where $Q = K = V = \mathbf{X}_{\text{pos}}$, and $W_i^Q, W_i^K, W_i^V \in \mathbb{R}^{F_p \times d_k}$, $W^O \in \mathbb{R}^{N_p d_k \times F_p}$ are learnable projection matrices and $d_k = \frac{K}{H}$. Each scaled dot-product attention head is computed as:

$$\text{Attention}(Q, K, V) = \text{softmax}\left(\frac{QK^\top}{\sqrt{d_k}} + \mathbf{M}^{Or}\right) V. \tag{38}$$

The resulting output $\mathbf{y} \in \mathbb{R}^{N_p \times F_p}$ of MultiHead$(Q, K, V)$ contains context-aware representations for each token.

### A.7.2 GUIDANCE OF LATENT DIFFUSION

We incorporate missing mask $\mathbf{M}^{or}$ and position encoding representations into intermediate layers of latent diffusion via a cross-attention mechanism, which has proven effective in aligning multi-modal signals such as language, image, and time-series features. Specifically, to process the input time series $\mathbf{X}^{Co}$, we first apply a positional encoder followed by a multihead attention encoder A.7.1 that outputs a refined representation $\mathbf{y} \in \mathbb{R}^{N_p \times F_p}$. We introduce a domain-specific encoder $\tau_\theta$ Rombach

et al. (2022) that projects $\mathbf{y}$ into an intermediate representation $\tau_\theta(\mathbf{y}) \in \mathbb{R}^{D \times 1}$. This encoded representation serves as the conditioning input to the model and is integrated into intermediate layers via a cross-attention mechanism. Specifically, for a given layer $i$, the cross attention is computed as:

$$\text{Attention}(Q, K, V) = \text{softmax}\left(\frac{QK^\top}{\sqrt{D}}\right) V,$$
$$Q = W_Q^{(i)} \cdot \varphi_i(\mathbf{h}),$$
$$K = W_K^{(i)} \cdot \tau_\theta(\mathbf{y}),$$
$$V = W_V^{(i)} \cdot \tau_\theta(\mathbf{y}). \tag{39}$$

Here, $\varphi_i(\mathbf{h}) \in \mathbb{R}^{D_i \times 1}$ denotes the flattened latent feature representation of the $i$-th layer, and $W_Q^{(i)} \in \mathbb{R}^{D \times D_i}$, $W_K^{(i)}, W_V^{(i)} \in \mathbb{R}^{D \times D}$ are learnable projection matrices. By integrating $\tau_\theta(\mathbf{y})$ into the model via attention, we enable fine-grained and dynamic conditioning on external guidance throughout the denoising process.

### A.7.3 CONTINUOUS ODE SAMPLING STRATEGY

To mitigate the stochasticity inherent in the latent sampling process, we adopt deterministic probability flow ODE sampling. The corresponding deterministic trajectory is governed by the following ordinary differential equation (ODE) Song et al.:

$$d\mathbf{x} = [\mathbf{f}(x, t) - \frac{1}{2}g(t)^2 \nabla_\mathbf{x} \log p_t(\mathbf{x})]dt \tag{40}$$

where $f(x, t) = \frac{1}{2}\beta(t)$ and $g(t) = \sqrt{\beta(t)}$ and we will obtain the sampling equation as:

$$d(\hat{\mathbf{Z}}_K)_t = \left[ \frac{\beta(t)}{2}(\hat{\mathbf{Z}}_K)_t + \frac{\beta(t)}{2} \frac{\boldsymbol{\epsilon}_\theta\big((\hat{\mathbf{Z}}_K)_t | \mathbf{X}_0^{Co}, t\big)}{\sqrt{1 - e^{(-\int \beta(t)dt)}}} \right] dt \tag{41}$$

Where $e^{(-\int \beta(t)dt)} = e^{\left(-\beta_{\text{start}}t - \frac{1}{2}(\beta_{\text{end}} - \beta_{\text{start}})t^2\right)}$.

### A.8 ALGORITHM DETAIL

The training algorithm is shown in Algorithm 1 while the sampling Algorithm is in Algorithm 2.

### A.9 DATASETS DETAIL

1. **PhysioNet 2012 Mortality Prediction Challenge (P2012) Silva et al. (2012)**: The PhysioNet 2012 Mortality Prediction Challenge (P2012) dataset comprises multivariate clinical time series collected from 4,000 ICU patients during the first 48 hours of admission. Each patient record includes 35 physiological and laboratory measurements sampled at irregular intervals. The dataset is highly sparse, with 80.52% of original missing values. The data is split and preprocessed as Tashiro et al. (2021).

2. **MIMIC-IV v3.1 Johnson et al. (2024)**: MIMIC-IV v3.1, released in October 2024, includes electronic health records from 364,627 patients admitted to the Beth Israel Deaconess Medical Center between 2008 and 2022. Following the preprocessing procedure in Harutyunyan et al. (2019), we retain eight vital signs—Diastolic blood pressure (BP), Fraction of inspired oxygen, Glucose, Heart rate, Mean BP, Oxygen saturation, Respiratory rate, and Systolic BP. Patients with fewer than 48 time steps are excluded to maintain tensor alignment. The final dataset comprises 36,401 patients, each with 48 time steps and 8 variables, with an overall original missing rate of 49.09%. The data is split and preprocessed as Hayat et al. (2022).

3. **Electricity Transformer Temperature (ETT) Zhou et al. (2021)**: The ETT dataset contains 15-minute interval readings from electricity transformers between July 1, 2016 and June 26, 2018, totaling 69,680 samples without original missing data. Each sample includes seven features: one oil temperature and six power load variables. The data is split and preprocessed as Du et al. (2023).

---

**Algorithm 1** Training of HSGM

---

1: **Input:** Time series values $\mathbf{X}_0^{Co}$, $\bar{\mathbf{X}}_0^{Or}$, $\mathbf{M}^{Ta}$ and $\mathbf{M}^{Or}$
2: **Output:** Latent varible $\mathbf{Z}_K$, Reconstructed original missing data $\hat{\mathbf{X}}_0^{Or}$ and model parameters $\lambda = \{\phi, \psi, \theta\}$, where $\phi$ and $\psi$ are the encoder, decoder and planar flow learnable parameters of VAE and normalizing flow while $\theta$ is the latent and observation diffusion learnable parameters.
3: Pre-train the NVAE architecture, optimize the Objective function Eq. 10 inputting $\mathbf{X}_0^{Co}$, $\bar{\mathbf{X}}_0^{Or}$
4: Frozen the encoder parameters $\phi$ of NVAE, obtain the latent distribution $\mathbf{Z}_K$
5: Initialize variables $\theta$ for latent diffusion
6: **for** each epoch in latent training **do**
7:     Add noise to $\mathbf{Z}_K$ by Eq. 11
8:     Optimize the latent objective function in Eq. 12 by taking gradient step on
9: $\nabla_\theta \|\boldsymbol{\epsilon} - \boldsymbol{\epsilon}_\theta((\mathbf{Z}_K)_t | \mathbf{X}_0^{Co}, t)\|^2$
10: **end for**
11: Obtain the reconstructed latent variable $\hat{\mathbf{Z}}_K$ from Algorithm 2
12: Calculate the cross attention Eq. 41 by inputting $\mathbf{Z}_K$ and $\hat{\mathbf{Z}}_K$
13: Post-train the decoder parameters $\psi$ of NVAE with the output of cross attention by $\left\| (\hat{\mathbf{X}} - \mathbf{X}_0) \odot \mathbf{M}^{Or} \right\|^2$
14: Obtain the reconstructed original missing data $\hat{\mathbf{X}}_0^{Or}$ by the trained decoder of VAE.
15: Initialize the variables $\theta$ for observation diffusion
16: **for** each epoch in observation diffusion **do**
17:     Add noise to $\mathbf{X}_0^{Ta}$
18:     Optimize the observation objective function in Eq. 30 by taking gradient step on
19: $\nabla_\theta \left\| \left( \boldsymbol{\epsilon} - \boldsymbol{\epsilon}_\theta(\mathbf{X}_t^{Ta}, t | \mathbf{X}_0^{Co}, \hat{\mathbf{X}}_0^{Or}) \right) \odot (1 - \mathbf{M}^{Ta}) \right\|^2$
20: **end for**
21: **return** Latent varible $\mathbf{Z}_K$, Reconstructed original missing data $\hat{\mathbf{X}}_0^{Or}$ and $\lambda$

---

**Algorithm 2** Sampling (Imputation) of HSGM

---

1: **Input:** Time series values $\mathbf{X}_0^{Co}$, $\mathbf{M}^{Ta}$, $\mathbf{M}^{Or}$, Latent varible $\mathbf{Z}_K$, Reconstructed original missing data $\hat{\mathbf{X}}_0^{Or}$ and $\lambda = \{\phi, \psi, \theta\}$
2: **Output:** The predicted missing values $\hat{\mathbf{X}}_0^{ta}$ and reconstructed latent Variable $\hat{\mathbf{Z}}_K$
3: Generate the Gassian Noise $\mathbf{Z}_T \sim \mathcal{N}(0, \mathbf{I})$
4: **for** $t = T$ to 1 in the latent space **do**
5:     Sample $(\hat{\mathbf{Z}}_K)_{t-1}$ using Eq. 41 with condition on $\mathbf{X}_0^{Co}$
6: **end for**
7: Obtain the reconstructed latent Variable $\hat{\mathbf{Z}}_K$
8: Generate the Gassian Noise $\mathbf{X}_T^{Ta} \sim \mathcal{N}(0, \mathbf{I})$
9: **for** $t = T$ to 1 in the observation space **do**
10:     Sample $\hat{\mathbf{X}}_{t-1}^{Ta}$ using Eq. 5 with condition on $\mathbf{X}_0^{Co}$ and $\hat{\mathbf{X}}_0^{Or}$
11: **end for**
12: Obtain the reconstructed simulated missing values $\hat{\mathbf{X}}_0^{Ta}$
13: **return** $\hat{\mathbf{Z}}_K$ and $\hat{\mathbf{X}}_0^{ta}$

---

4. **Synthetic dataset Fang et al. (2024)**: A synthetic dataset with highly spatio-temporal correlations of 4 channels, and each channel is a mixture of multiscale trend and seasonality factors. The dataset can be generated by 4 correlation functions and a weight matrix, which are defined as:

$$\mathbf{S}(t) = \frac{\mathbf{U}\mathbf{V}(t)}{10}, \quad \text{Where } \mathbf{U} = \begin{pmatrix} 1 & 1 & -2 & -2 \\ 0.4 & 1 & 2 & -1 \\ -0.3 & 2 & 1 & 1 \\ -1 & 1 & 1 & 0.5 \end{pmatrix}, \quad \mathbf{V}(t) = \begin{pmatrix} 10t \\ \sin(20\pi t) \\ \cos(40\pi t) \\ \sin(60\pi t) \end{pmatrix}. \tag{42}$$

2000 data points over 500 are irregularly sampled timestamps from $[0, 1]$ as the same as Fang et al. (2024). The dataset is divided into training (70%), validation (10%), and test (20%) sets.

## A.10 EXPERIMENTAL SETTINGS

We implement a VAE and a normalizing flow with NVAE Vahdat & Kautz (2020), and use CSDI for observation diffusion Tashiro et al. (2021). The normalizing flow uses $K = 4$ transformations. For latent diffusion, the batch size is 8 and we use a linear noise schedule with $\beta_{\text{start}} = 0.1$ and $\beta_{\text{end}} = 20$, adopting the NCSN architecture Song et al. to learn the score based function. To reduce randomness, we draw 100 samples of $\hat{\mathbf{Z}}$ and report their mean as the final output. Latent ODE sampling in Eq. 41 is performed with a continuous ODE solver Chen et al. (2018). All experiments are run in PyTorch 1.13.1 on a Linux server with an Intel Core i7 1800H at 2.30 GHz, an NVIDIA GeForce RTX 3080, and 32 GB memory.

## A.11 METRICS

To evaluate the imputation performance of different methods, we adopt mean absolute error (MAE) and root mean square error (RMSE) as:

- Mean Absolute Error(MAE):

$$MAE(\mathbf{X}^{Ta}, \hat{\mathbf{X}}^{Ta}) = \frac{\left\| (\mathbf{X}^{Ta} - \hat{\mathbf{X}}^{Ta}) \odot (1 - \mathbf{M}^{Ta}) \right\|_1}{\|1 - \mathbf{M}^{Ta}\|_1}$$

- Root Mean Squared Error(RMSE):

$$RMSE(\mathbf{X}^{Ta}, \hat{\mathbf{X}}^{Ta}) = \frac{\left\| (\mathbf{X}^{Ta} - \hat{\mathbf{X}}^{Ta}) \odot (1 - \mathbf{M}^{Ta}) \right\|}{\|1 - \mathbf{M}^{Ta}\|}$$

where $\|\bullet\|_1$ and $\|\bullet\|$ denotes L1 norm and L2 norm. $RMSE$ and $MAE$ are quantitatively used to describe the difference between the predictive value and the ground truth value. The smaller the value is, the more accurate the model is.

To evaluate the uncertainty of the generative model, we adopt the continuous ranked probability score (CRPS) Matheson & Winkler (1976) to evaluate the compatibility of the estimated probability distribution with the observed value. For a missing value $x$ whose estimated probability distribution is $D$, CRPS measures the compatibility of $D$ and $x$, which can be defined as the integral of the quantile loss $\Lambda_\alpha$:

$$\text{CRPS}(D^{-1}, x) = \int_0^1 2\Lambda_\alpha(D^{-1}(\alpha), x)d\alpha, \tag{43}$$

$$\Lambda_\alpha(D^{-1}(\alpha), x) = (\alpha - \mathbb{I}_{x < D^{-1}(\alpha)})(x - D^{-1}(\alpha)), \tag{44}$$

where $\alpha \in [0, 1]$ is the quantile level, $D^{-1}(\alpha)$ is the $\alpha$-quantile of distribution $D$, and $\mathbb{I}$ is the indicator function. Since our distribution of missing values is approximated by generating 100 samples, we compute quantile losses for discretized quantile levels with 0.05 ticks following Tashiro et al. (2021) as:

$$\text{CRPS}(D^{-1}, x) \simeq \frac{1}{19} \sum_{i=1}^{19} 2\Lambda_{i \times 0.05}(D^{-1}(i \times 0.05), x). \tag{45}$$

We compute CRPS for each estimated missing value and use the average as the evaluation metric, which is formalized as:

$$\text{CRPS}(D, \mathbf{X}^{ta}) = \frac{1}{|\mathbf{X}^{ta}|} \sum_{x \in \mathbf{X}^{ta}} \text{CRPS}(D^{-1}, x). \tag{46}$$

The smaller the CRPS value, the less uncertainty there is in the imputation result.

### A.12 BASELINES

We compare the performance of HSGM against a diverse set of baseline methods, including traditional statistical approaches, matrix factorization techniques, and recent deep learning-based models:

- **Mean**: A naive baseline that fills missing values using the mean of each node over the entire time horizon.

- **KNN**: Estimates missing values by averaging the values of the 3 nearest neighboring nodes.

- **MICE** Van Buuren (2000): Conducts multiple imputations through chained equations; we set the maximum iterations to 100.

- **MF** (Cichocki & Phan, 2009): Performs matrix completion via singular value decomposition (SVD) to recover missing entries from low-rank structure.

- **M$^2$DMTF** (Fan, 2021): Performs imputation using multi-mode deep matrix and tensor factorization.

- **Transformer** (Vaswani et al., 2017): Applies a multi-head attention mechanism for capturing long-range dependencies in time series imputation.

- **BRITS** (Cao et al., 2018): Utilizes a bidirectional RNN structure to iteratively infer missing values.

- **SAITS** (Du et al., 2023): Employs a self-attention-based architecture tailored for time series imputation under a self-supervised setting.

- **MPGRU** (Li et al., 2018): Integrates graph neural networks with GRU for spatio-temporal imputation.

- **GRIN** (Cini et al., 2021): Combines GNN and bidirectional GRU in a two-stage framework for structured time series imputation.

- **HSPGNN** (Liang et al., 2024): Use the physics-incorporated neural network with attention and GNN for imputation.

- **CSDI** (Tashiro et al., 2021): Leverages conditional score-based diffusion models for time series imputation, explicitly trained to model correlations in observed data, achieving strong performance on healthcare and environmental datasets.

- **FGTI** Yang et al. (2024) integrates frequency-domain information into a diffusion model for multivariate time-series imputation, emphasizing residual terms via high-frequency filtering and complementing trend and seasonal components through dominant-frequency filtering.

- **BayOTIDE** Fang et al. (2024) is a Bayesian model for online multivariate time series imputation, decomposing the series into a temporal function basis and channel-wise weights modeled with Gaussian processes (GPs). An efficient online inference algorithm leverages the SDE representation of GPs and moment-matching.

- **LSSDM** (Liang et al., 2025): Performs unsupervised time series imputation by learning a low-dimensional latent representation of observed data and refining coarse reconstructions via conditional diffusion, enabling high-fidelity imputation with uncertainty estimation.

- **DiffPuter** Zhang et al. (2025) combines diffusion models with the Expectation-Maximization algorithm to address missing data imputation. It iteratively learns the joint distribution of observed and missing values and performs conditional sampling.

### A.13 Bias Visualization of Score-Based Functions on the ETT Dataset

We conduct experiments on the ETT dataset to evaluate the biased behavior of score-based functions.

- **Bias and accumulated bias.** We report the MAE of both bias and accumulated bias, as illustrated in Fig. 6.

- **Heat map visualization.** We visualize the score-based functions as heat maps across different reverse time steps. Fig. 7 presents the ground truth, HSGM, and CSDI results at reverse steps 1, 40, and 49.

### A.14 Ablation Study

To evaluate the contribution of each component in HSGM, we conduct an ablation study, with results shown in Tab. 3. The results indicate that a VAE without normalizing flows is limited in capturing complex latent distributions, whereas normalizing flows provide a more flexible latent representation. Furthermore, unifying the latent diffusion and observation diffusion leads to improved imputation performance, especially when the dataset contains a high proportion of original missing values. On the ETT dataset, latent diffusion alone yields superior imputation performance compared to the observation diffusion layer. However, incorporating the output of latent diffusion consistently enhances the performance of the observation diffusion layer for all datasets.

Table 3: Performance comparison across datasets and components

| Model | P2012@50% | | MIMIC IV@50% | | ETT@Block Missing | | Synthetic@50% | |
|---|---|---|---|---|---|---|---|---|
| | MAE | RMSE | MAE | RMSE | MAE | RMSE | MAE | RMSE |
| VAE-Non norm | 0.382±0.003 | 0.624±0.016 | 0.046±0.002 | 0.132±0.003 | 0.207±0.002 | 0.311±0.003 | 0.198±0.010 | 0.248±0.010 |
| VAE-norm | 0.374±0.003 | 0.610±0.015 | 0.045±0.002 | 0.129±0.003 | 0.190±0.002 | 0.294±0.004 | 0.190±0.010 | 0.234±0.012 |
| Latent Diffusion | 0.329±0.003 | 0.569±0.015 | 0.041±0.002 | 0.122±0.003 | **0.180±0.002** | **0.289±0.004** | 0.173±0.010 | 0.215±0.014 |
| Observation Diffusion(CSDI) | 0.301±0.002 | 0.614±0.017 | 0.050±0.001 | 0.178±0.002 | 0.227±0.004 | 0.606±0.005 | 0.136±0.011 | 0.204±0.012 |
| Latent+Observation Diffusion | **0.241±0.003** | **0.538±0.015** | **0.032±0.002** | **0.109±0.003** | 0.220±0.004 | 0.581±0.005 | **0.104±0.010** | **0.146±0.011** |

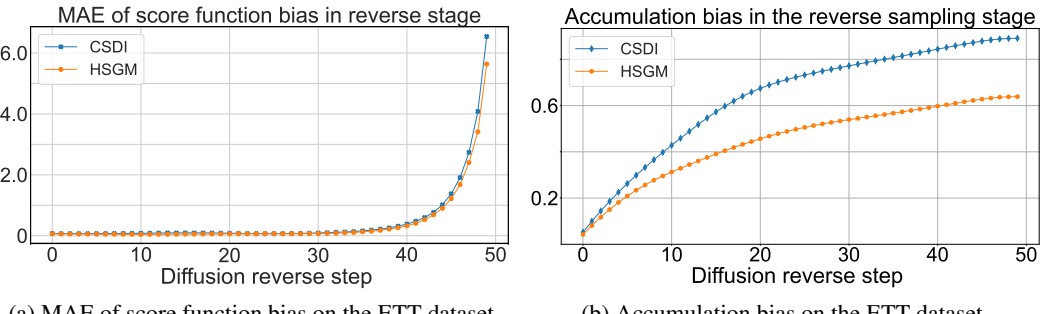

(a) MAE of score function bias on the ETT dataset.    (b) Accumulation bias on the ETT dataset.

Figure 6: MAE comparison of bias and accumulated bias on the ETT dataset.

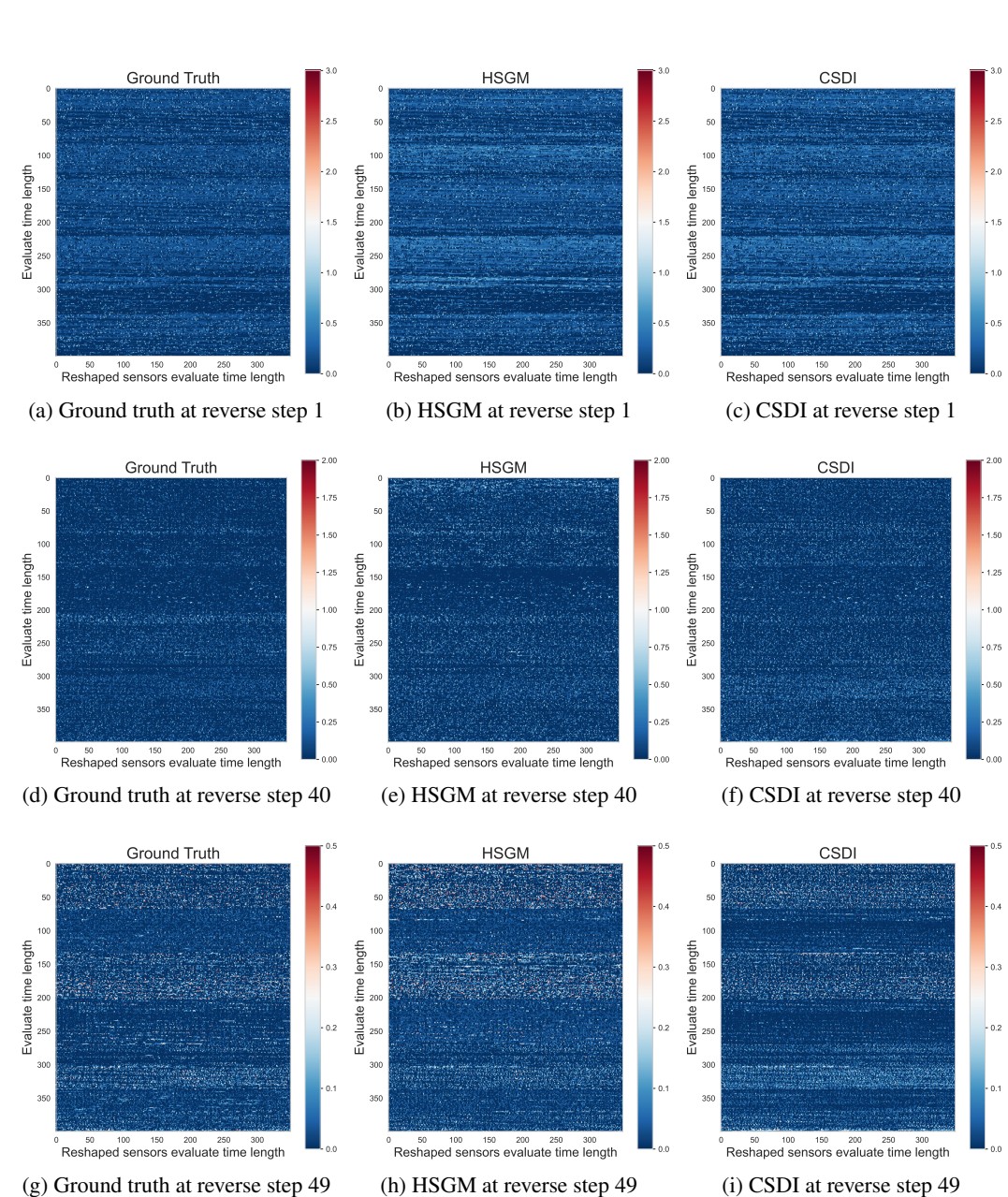

Figure 7: Heat map visualization of score-based functions at different reverse time steps on the ETT dataset.

