# OpenReview forum: "Latent-to-Observable Score Correction for Probabilistic Time Series Imputation"
_ICLR.cc/2026/Conference — Submitted to ICLR 2026_

### Official Review · Reviewer_dwA9 · 2025-10-16

**Soundness:** 3
**Presentation:** 2
**Contribution:** 2
**Rating:** 4
**Confidence:** 5

**Summary:**

To address the issue that existing score-based generative models ignore originally missing data during training, which leads to biased score estimation, this paper first provides a theoretical analysis of the impact of missingness under the DDPM framework. It then proposes HSGM, which integrates latent-space and observation-space diffusion within a layer-wise refinement framework grounded in the chain rule of probability. This design enables the model to handle originally missing data without requiring ground-truth supervision during training. Finally, extensive experiments are conducted to validate the effectiveness of HSGM.

**Strengths:**

- S1：This paper provides a comprehensive theoretical analysis.
- S2：The paper presents extensive experimental results and compares HSGM with a wide range of baseline methods.

**Weaknesses:**

- W1: The motivation is not sufficiently clear. It remains unclear why existing score-based models would perform worse under scenarios with high missing rates and strong inter-variable correlations when original missing data are ignored. It would be helpful to include experiments comparing the performance of existing score-based models with and without original missing data, particularly under datasets characterized by high missingness and strong inter-variable correlations. Such experiments would better clarify the motivation of HSGM.
- W2：The experimental setup is not clearly described. The settings of the ablation study variants are insufficiently specified, and there is no reasonable analysis explaining why Latent Diffusion or VAE performs better on the ETT dataset. In addition, more detailed ablation variants are needed. For example, the paper emphasizes the importance of the cross-attention mechanism, but no corresponding ablation variant is provided to support this conclusion.
- W3：The experimental analysis is incomplete. For instance, on the P2012 and MIMIC-IV datasets, HSGM shows significantly worse CRPS scores compared to the baselines. It would be important to analyze whether this is caused by certain characteristics of the datasets or other factors.
- W4：The model complexity is concerning. HSGM involves two diffusion stages and incorporates a cross-attention mechanism, yet lacks any analysis of computational complexity, model parameter count, runtime efficiency, or hyperparameter sensitivity. Such analyses are essential for this work.
- W5：The paper lacks necessary explanations of some notations, such as the meaning of N and F in line 136. There are also minor typographical issues, such as the misspelling of “synthetic” in Figure 3.

**Questions:**

- Q1：How does the performance of HSGM change with varying missing rates, and how does its robustness compare with the baselines?

---

> ### Author Response · Authors · 2025-11-21
> **Response to Reviewer dwA9 (Part I)**
>
> $\textbf{W1}$: Please refer to our response to Reviewer niE6 for the detailed answer to W1. The P2012 and MIMIC datasets contain a substantial amount of original missing data, whereas the ETT and Synthetic datasets contain none. For the ETT and Synthetic datasets, we treat the simulated missing values as original missing data: they are excluded during training but used for the final evaluation. All of these datasets have been shown to exhibit strong inter-variable correlations. The results are presented below, where w/o original missing indicates that simulated missing data are allowed to participate in training, while w/ original missing indicates that they are not. The results further validate the effectiveness of our model.
>
> **Table: Comparison of imputation performance on ETT and Synthetic datasets**
>
> | Dataset                       | CSDI MAE | CSDI RMSE | FGTI MAE | FGTI RMSE | LSSDM MAE | LSSDM RMSE | DiffPuter MAE | DiffPuter RMSE | HSGM MAE | HSGM RMSE |
> |-------------------------------|----------|-----------|----------|-----------|-----------|------------|----------------|-----------------|----------|-----------|
> | ETT w/o original missing      | 0.220    | 0.576     | 0.219    | 0.411     | 0.220     | 0.576      | 0.416          | 0.815           | 0.172    | 0.263     |
> | ETT w/ original missing       | 0.227    | 0.606     | 0.225    | 0.418     | 0.221     | 0.585      | 0.605          | 1.073           | 0.180    | 0.289     |
> | Synthetic w/o original missing| 0.065    | 0.093     | 0.084    | 0.114     | 0.065     | 0.093      | 0.091          | 0.117           | 0.065    | 0.093     |
> | Synthetic w/ original missing | 0.136    | 0.204     | 0.143    | 0.188     | 0.112     | 0.157      | 0.133          | 0.199           | 0.104    | 0.146     |
>
>
> $\textbf{W2}$: Below we clarify the ablation configurations.
>
> - VAE-NonNorm: A standard VAE without normalizing flow, assuming a simple Gaussian latent distribution.
>
> - VAE-Norm: A VAE equipped with a normalizing flow, enabling a more expressive non-Gaussian latent distribution.
>
> - Latent Diffusion: Uses the pretrained VAE-Norm and applies latent diffusion with the proposed cross-attention module.
>
> - Observation Diffusion: Replaces original missing values with zeros and trains an observation-space diffusion model analogous to CSDI.
>
> - Latent + Observation Diffusion: Samples imputed original missing values using latent diffusion and forwards them into observation-space diffusion.
>
> For the ETT and synthetic datasets, our setup treats simulated missing data as original missing values: they are excluded from the training stage but used for evaluation in the test stage. This differs from the P2012 and MIMIC datasets, which contain both original and simulated missing entries. Under this setting, latent diffusion and VAE sometimes outperform observation diffusion due to their unsupervised nature, especially when labels for original missing data are unavailable during training. However, this is not universally guaranteed; performance depends on dataset-specific characteristics such as inter-variable correlation strength and variable importance.
>
> Regarding the cross-attention mechanism, we provide a detailed discussion and quantitative evaluation in the Generation vs. Reconstruction section of the experiments. The effectiveness of cross-attention is demonstrated in Table 2 across multiple settings, which reports its contribution to performance improvements.
>
> $\textbf{W3}$: Thank you for the insightful question. The higher CRPS of HSGM on P2012 and MIMIC-IV is due to the nature of CRPS, which measures probabilistic calibration rather than pointwise accuracy. Some baseline models produce overly wide or over-smoothed predictive distributions, leading to poor MAE/MSE but artificially lower CRPS, whereas HSGM generates sharper and more accurate predictions, as shown in Fig 3 (c,d,g,h), achieving significantly better MAE/MSE but slightly higher CRPS if the predictive distribution is not perfectly calibrated. This effect is amplified by dataset properties: P2012 is highly sparse and irregular, while MIMIC-IV contains heterogeneous trajectories, both making probabilistic calibration challenging.
>
> $\textbf{W4}$: Please refer to our response to W3 from Reviewer xT6t for a detailed discussion on time complexity. Regarding hyperparameter settings, some are listed in the Experimental Settings section, as we conducted experiments to select the optimal values, while others follow well-established configurations from cited works such as CSDI and NVAE. For the number of model parameters, please refer to the supplemental code, as it varies depending on the dataset.
>
> $\textbf{W5}$: Thank you for pointing this out. Here, $N$ denotes the total length of the time series, and $F$ denotes the total number of sensors. We have updated the manuscript accordingly and corrected the typographical errors, including the misspelling of “synthetic” in Figure 3.

---

> ### Author Response · Authors · 2025-11-21
> **Response to Reviewer dwA9  (Part II)**
>
> $\textbf{Q1}$: To evaluate the robustness of HSGM compared with the baselines, we conducted experiments on  Synthetic datasets, where we can control the original missing rate. Also, we treat the simulated missing values as original missing data: they are excluded during training but used for the final evaluation. The results are summarized below:
>
> **Table: Imputation performance on Synthetic dataset under different missing ratios**
>
> | Method      | 10% MAE | 10% RMSE | 20% MAE | 20% RMSE | 30% MAE | 30% RMSE | 40% MAE | 40% RMSE | 50% MAE | 50% RMSE |
> |-------------|---------|----------|---------|----------|---------|----------|---------|----------|---------|----------|
> | CSDI        | 0.049  | 0.070   | 0.056  | 0.075   | 0.080  | 0.140   | 0.099  | 0.156   | 0.136  | 0.204   |
> | FGTI        | 0.051  | 0.073   | 0.058  | 0.083   | 0.095  | 0.152   | 0.102  | 0.159   | 0.143  | 0.188   |
> | LSSDM       | 0.038  | 0.059| 0.051  | 0.073   | 0.071  | 0.102   | 0.086  | 0.126   | 0.112  | 0.157   |
> | DiffPuter   | 0.058  | 0.082   | 0.063  | 0.081   | 0.085  | 0.123   | 0.110  | 0.153   | 0.133  | 0.199   |
> | HSGM (ours) | 0.034  | 0.051   | 0.045  | 0.064   | 0.059  | 0.087   | 0.074  | 0.110   | 0.104 | 0.146   |

---

> > ### Comment · Reviewer_dwA9 · 2025-11-21
> >
> > Thank you for your responses. I still have some questions:
> >
> > You mentioned that “we treat the simulated missing values as original missing data: they are excluded during training but used for the final evaluation.” Since you clarify that initializing original missing values with zeros is problematic for CSDI, may I ask how you initialize or represent the original missing positions during training in your method?
> >
> > In addition, how can we demonstrate that under strong inter-variable correlations, including original missing values leads to worse performance? How do you quantify the degree of inter-variable correlations in the dataset?
> >
> > You should also consider adding results under higher missing rates, such as 60–70%. In your paper, you mentioned that under high missing rates, ignoring original missing values can significantly distort the learned score function even at non-missing points. Would this imply that your method might exhibit a more pronounced advantage under higher missing ratios? Moreover, I noticed a possible trend that HSGM may deteriorate more quickly than LSSDM as the missing rate increases.

---

> > > ### Author Response · Authors · 2025-11-26
> > > **Response to Reviewer dwA9 (Part Ⅲ)**
> > >
> > > $\textbf{Answer 1}$: Our model explicitly accounts for missing positions using a cross-attention mechanism equipped with both a missing-value mask and positional encoding within the latent diffusion module. This is illustrated in the latent diffusion block in Figure 2. Additional implementation details are provided in Appendix A.7.1 and A.7.2.
> > >
> > > $\textbf{Answer 2}$: Thank you for your questions. To address this concern, we designed an additional experiment to illustrate how strong inter-variable correlations and original missing values can affect imputation performance. As described in Appendix A.9 (Equation 42), our synthetic data are generated via a linear mixing process
> > > $
> > > X(t) = U V(t),
> > > $
> > > which naturally induces cross-channel correlations controlled by the choice of mixing matrix $U$. For this experiment, we constructed the following matrices:
> > >
> > > $U = [(1, 1, -2); (1.05, 0.95, -2.10); (1, -1, 0)] $
> > > $
> > > V(t) =
> > > [
> > > \sin(20\pi t) ;
> > > \cos(40\pi t) ;
> > > \sin(60\pi t)
> > > ]$
> > >
> > > From the row vectors of $U$, it is clear that $\textbf{row 1 and row 2 are highly similar}$, indicating strong correlations between the corresponding channels of $X(t)$. In contrast, $\textbf{row 1 and row 3 are nearly orthogonal}$, implying weak correlations.
> > >
> > > We generated 500 time points over $[0,1]$ and computed the Pearson correlation matrix of $X(t)$:
> > > $\[
> > > (1 , 0.9993961 , -0.02268837);
> > > (0.9993961 , 1 , 0.00483626);
> > > (-0.02268837, 0.00483626,1) \]$
> > >
> > > This confirms that channels $X_1(t)$ and $X_2(t)$ are $\textbf{strongly correlated}$ (correlation $\approx 1$), whereas $X_3(t)$ is $\textbf{weakly correlated}$ with both ( correlation $\approx 0$).
> > >
> > > Next, we removed 50% of the values in channel $X_1(t)$ while keeping all values in $X_2(t)$ and $X_3(t)$ available for training. We conducted experiments under two settings: in the first, all values in channel $X_1(t)$ are used for training and inference, which we denote as CSDI without original missing; in the second, the missing values in channel $X_1(t)$ are excluded from both training and inference, which we denote as CSDI with original missing. The results are shown as follows:
> > >
> > > |                                   |   X₂    |        |   X₃    |        |
> > > |:---------------------------------:|:-------:|:------:|:-------:|:------:|
> > > |                                   |   MAE   |  RMSE  |   MAE   |  RMSE  |
> > > | CSDI with original missing    | 0.1104  | 0.2991 | 0.0439  | 0.1107 |
> > > | CSDI without original missing | 0.0141  | 0.0259 | 0.0294  | 0.0921 |
> > > | Improved                     | 0.0963  | 0.2732 | 0.0145  | 0.0186 |
> > > | Improved percentage          | 87.2%   | 91.3%  | 33.0%   | 16.8%  |
> > >
> > >
> > > The results indicate that missing values in a strongly correlated channel (here, $X_1$) substantially degrade the performance of its correlated counterpart $X_2$, even though $X_2$ has no missing values during training, while the weakly correlated channel $X_3$ is minimally affected. This observation directly supports Proposition 4.2.
> > >
> > > $\textbf{Answer 3}$: LSSDM can be regarded as an approximation of our method, as it relies on a standard Gaussian VAE with limited representational capacity. The Gaussian assumption in the VAE latent space limits its expressiveness, resulting in less accurate imputations compared to latent diffusion. Across all datasets, our model achieves more accurate imputations than LSSDM, as evidenced by both Table 1 and Table 2. Also, We conduct imputations at different missing rates. The results are summarized in the following table:
> > >
> > > |            |         |       |       |       |       |       |       |    Synthetic |     dataset     |       |       |       |       |       |       |       |       |       |
> > > |:----------:|:-----:|:-----:|:-----:|:-----:|:-----:|:-----:|:-----:|:-----:|:-----:|:-----:|:-----:|:-----:|:-----:|:-----:|:-----:|:-----:|:-----:|:-----:|
> > > |            |  10%  |       |  20%  |       |  30%  |       |  40%  |       |  50%  |       |  60%  |       |  70%  |       |  80%  |       |  90%  |       |
> > > |            |  MAE  |  RMSE |  MAE  |  RMSE |  MAE  |  RMSE |  MAE  |  RMSE |  MAE  |  RMSE |  MAE  |  RMSE |  MAE  |  RMSE |  MAE  |  RMSE |  MAE  |  RMSE |
> > > |    CSDI    | 0.049 | 0.070 | 0.056 | 0.075 | 0.080 | 0.140 | 0.099 | 0.156 | 0.136 | 0.204 | 0.177 | 0.242 | 0.218 | 0.304 | 0.276 | 0.363 | 0.406 | 0.475 |
> > > |    FGTI    | 0.051 | 0.073 | 0.058 | 0.083 | 0.095 | 0.152 | 0.102 | 0.159 | 0.143 | 0.188 | 0.187 | 0.272 | 0.228 | 0.312 | 0.292 | 0.385 | 0.426 | 0.502 |
> > > |    LSSDM   | 0.038 | 0.059 | 0.051 | 0.073 | 0.071 | 0.102 | 0.086 | 0.126 | 0.112 | 0.157 | 0.124 | 0.201 | 0.189 | 0.254 | 0.224 | 0.283 | 0.291 | 0.402 |
> > > |  DiffPuter | 0.058 | 0.082 | 0.063 | 0.081 | 0.085 | 0.123 | 0.110 | 0.153 | 0.133 | 0.199 | 0.176 | 0.223 | 0.196 | 0.262 | 0.243 | 0.345 | 0.294 | 0.428 |
> > > | HSGM(ours) | 0.034 | 0.051 | 0.045 | 0.064 | 0.059 | 0.087 | 0.074 | 0.110 | 0.104 | 0.146 | 0.120 | 0.189 | 0.157 | 0.233 | 0.197 | 0.263 | 0.247 | 0.344 |

---

### Official Review · Reviewer_niE6 · 2025-10-29

**Soundness:** 2
**Presentation:** 3
**Contribution:** 2
**Rating:** 2
**Confidence:** 3

**Summary:**

This paper proposes Hierarchical Score-Based Generative Model (HSGM). Authors first observe that if we run DDPM on imputing missing data with zero or mean, there is intrinsic bias, which basically originates from the discrepancy between noisy distribution (probability path) seen during training and during inference. To mitigate this, HSGM propose to use another latent diffusion model that first imputes missing data, then run original generative model.

**Strengths:**

- The flow of the paper is great. They first provide theoretical insight, then propose the method that address the problem, and show that empirically it works.
- Empirical results seems good.

**Weaknesses:**

- I think proposition 4.1 and 4.2 is inappropriate for this context. 4.1 and 4.2 assumes that the original DDPM model is trained without missing values. These propositions are saying that if we train our model without missing values during training, but then if we have missing values during inference, there is a significant bias. This is totally true, but not interesting, because the condition information we give to the model is completely changed. This is not the case how imputation model is trained/used. Even CSDI paper used in this paper give missingness information to the model, so the model is trained to adapt its score function for missing values. If we add this additional condition to the model, then the analysis on section 4 should be completely re-written. (Correct me if I am misunderstanding)

**Questions:**

- Please address weakness. Since theoretical analysis is crucial point of this paper, if this part is addressed, I will re-evaluate and increase the score.
- Do we actually need expensive diffusion model to impute missing values? Please compare with using CSDI (or other baselines) with lighter imputation methods

---

> ### Author Response · Authors · 2025-11-20
> **Response to Reviewer niE6**
>
> $\textbf{W1}$: We first clarify that the symbols $\mathbf{X}\_{t}^{\overline{\mathrm{Or}}}$ and $\mathbf{\bar{X}}^{\mathrm{Or}}$ are not the same. Their definitions are given in Lines 141 and 291: $\mathbf{X}\_{t}^{\overline{\mathrm{Or}}}$ denotes replacing the original missing entries with zeros, whereas $\mathbf{\bar{X}}^{\mathrm{Or}}$ denotes applying linear interpolation to the original missing data.
>
>  Propositions 4.1 and 4.2 do not assume that the original DDPM is trained without missing data. Rather, they provide a theoretical and quantitative analysis of the bias between the score function computed from complete data and the score function computed with original missing data. Specifically, Proposition 4.1 characterizes the bias for each individual entity, while Proposition 4.2 quantifies the bias for correlated entities. Importantly, due to the presence of original missing data, the true score function computed from complete datascore function is never obtained during either training or inference.
>
> I would like to clarify the distinction between original missing data and simulated missing data. The original missing data refers to values that are missing in the raw dataset, whereas simulated missing data are artificially masked for evaluation during the inference stage. During training, we allow the use of the ground truth for the simulated missing data.  In addition, CSDI focuses on simulated missing data, assumes that the original missing values are independent of the observed data, and replaces them with zeros due to the absence of labels.
>
> Also, $\mathbf{s}$ denotes the true score, $\mathbf{s}\_\theta$ is the score function learned by the neural network (e.g., U-Net), and $\mathbf{s}\_\theta *$ is the optimal score function learned by the network. Importantly, if the score function is biased due to original missing data, then the neural network can only learn a score estimator $\mathbf{s}\_\theta$ that approximates $\mathbf{s}(\mathbf{X}\_t^{\overline{Or}}, t)$ rather than the true score $\mathbf{s}(\mathbf{X}\_t, t)$. As shown in Eq. 5, this mismatch inevitably propagates to the inference stage and leads to biased imputation even there is no missing data in the test dataset.
> Because original missing data exist during training, $\mathbf{s}\_\theta *$ inherently contains bias in the mapping relationship. This bias is intrinsic to the learned score function and cannot be corrected simply by multi-sample averaging at inference.
>
>
> In many real-world domains, especially healthcare, the proportion of original missing data is substantial. Imputing these entries is considerably more challenging because the true values are inherently unknown. This fundamental difficulty motivates us to apply the latent diffusion, as shown in Eq. 9, to learn the original missing data in an unsupervised way to reduce this bias caused by original missing data, rather than simply replacing them with zeros as done in CSDI.
>
> $\textbf{Q1}$: Please refer to our response to W1.
>
> $\textbf{Q2}$:The necessity of using a expensive diffusion-based model for imputation depends on both the original missing rate and the correlations among different elements of the dataset. For original missing data, the missing rate is fixed and often uncontrollable, and the true values are usually unavailable.
>
> As indicated by Corollary 4, a computationally expensive diffusion model is not required when the original missing rate is low. However, in domains such as healthcare, missing rates are often very high because measurements cannot always be obtained from patients. Proposition 4.2 further suggests that when correlations between variables are weak, even a single-layer diffusion model can suffice.
>
> In practice, lighter imputation methods such as KNN, mean imputation, and matrix factorization can also be applied. Comparative results are provided in Table 1. If time complexity is your concern, please refer to our response to Reviewer xT6t regarding W3.

---

> > ### Author Response · Authors · 2025-11-26
> >
> > Regarding W1 and Q1, as inspired by Reviewer dwA9, we conducted the related experiments. The details and results are presented in Response to Reviewer dwA9 (Part Ⅲ) of Answer 2.

---

### Official Review · Reviewer_xT6t · 2025-10-30

**Soundness:** 2
**Presentation:** 3
**Contribution:** 2
**Rating:** 2
**Confidence:** 5

**Summary:**

This paper introduces a hierarchical score-based diffusion model for probabilistic time series imputation. In this model, a pre-trained VAE is used to model the latent distribution, and a cross-attention module is used to enhance the data generation performance.

**Strengths:**

1) Relevant derivations regarding the proposed method are given.
2) The idea of using a pre-trained VAE to learn the latent distribution for the diffusion process seems reasonable.
3) The idea of using cross-attention to improve the fidelity of the data sampling process seems feasible.

**Weaknesses:**

1)	Using a latent VAE and cross-attention to improve the diffusion model’s performance is not a very novel idea.
2)	Can the proposed method be accelerated like other types of diffusion models, e.g., DDIM and Rectified Flow?
3)	The computational and time costs of the proposed method compared to other diffusion models are unclear.
4)	Using multi-sample averages from different initial noise can mitigate sampling bias of diffusion models. The author should also discuss this approach and compare it with the proposed method.
5)	Minor: typo in Line 325, it should be “obtained”

**Questions:**

1)	The idea of debiasing does not make much sense as the sampling trajectories of the DDPM is stochastic. What is the unbiased estimation in the context？

2)	It’s not hard to prove that the bias is only related to the time-dependent variance as indicated in Eq (18). However, one of my main concerns is that the linear approximation in Eq (18) might not be accurate enough as the score function is learned from the data.

3)	To let Eq (6) hold, it has to assume that the denoising process is absolutely accurate for x_t (which holds for the forward process with a pre-defined noise scheduler), however the score estimated by the neural network is only an approximation. Therefore, I have some doubt about Eq (6).

4)	Can the proposed method be used for other data types besides time series as well?

---

> ### Author Response · Authors · 2025-11-20
> **Response to Reviewer xT6t  (Part I)**
>
> $\textbf{W1}$: Please refer to our response to Reviewer W2GG regarding comment W3.
>
> $\textbf{W2}$: Our method is built on standard diffusion model components. Consequently, it can be accelerated using established techniques such as DDIM or Rectified Flow without affecting the underlying theoretical framework.
>
> $\textbf{W3}$: The overall computational cost of our method consists of the observation DDPM stage plus the latent diffusion stage with probability flow. Specifically:
>
> - Observation DDPM: The time complexity is $O(T (N F)^2 C_1^{\text{unet}})$, where $T$ is the number of diffusion steps, $N$ and $F$ are the sequence length and feature dimensions, and $C_1^{\text{unet}}$ reflects the cost of the U-Net operations.
> - Latent diffusion with normalizing flow: The time complexity is $O(T_{\text{latent}} D^2 C_2^{\text{unet}})$, where $D$ is the latent dimension, $T_{\text{latent}}$ is the number of diffusion steps in the latent space, and $C_2^{\text{unet}}$ is the U-Net cost. The additional cross-attention contributes $O(D^2)$ complexity.
>
> Overall, the method’s cost scales linearly with the number of diffusion steps and quadratically with feature or latent dimensions, comparable to other diffusion-based imputation models.
>
> $\textbf{W4}$: The reviewer refers to mitigating sampling bias at the observation-space sampling stage. As shown in Eq. 5, using multiple samples from different initial noise and averaging them can reduce the bias introduced by stochastic noise during sampling.
>
> However, this approach does not address the bias in the learned score function itself, which arises during training due to original missing data. To clarify the notation: $\mathbf{s}$ denotes the true score, $\mathbf{s}\_\theta$ is the score function learned by the neural network (e.g., U-Net), and $\mathbf{s}\_\theta \*$ is the optimal score function learned by the network. Because original missing data exist during training, then the neural network can only learn a score estimator $\mathbf{s}\_\theta$ that approximates $\mathbf{s}(\mathbf{X}\_t^{\overline{Or}}, t)$ rather than the true score $\mathbf{s}(\mathbf{X}_t, t)$, $\mathbf{s}\_\theta^*$ inherently contains bias in the mapping relationship. This bias is intrinsic to the learned score function and cannot be corrected simply by multi-sample averaging at inference. Furthermore, in the sampling stage, the conditional input $\mathbf{X}\_t$ changes if the imputed original missing values are obtained from the latent diffusion. This means that the original missing data can affect both the mapping relationship of the learned score function and the input $\mathbf{X}\_t$ during the sampling stage, as shown in Eq. 5.
>
> In our method, multi-sample averaging is still applied at inference (Algorithm 1, 100 samples) to mitigate noise-related stochasticity, but the key contribution of our approach is addressing the structure bias in the score function, which standard multi-sample averaging cannot reduce. This fundamental difficulty motivates us to apply the latent diffusion to learn the original missing data in an unsupervised way to reduce this bias caused by original missing data, rather than simply replacing them with zeros as done in CSDI.
>
> $\textbf{W5}$: Thank you for your feedback. We have corrected the typo.

---

> ### Author Response · Authors · 2025-11-20
> **Response to Reviewer xT6t  (Part II)**
>
> $\textbf{Q1}$: Please refer to our responses to W4 and Q2 for details.
>
> $\textbf{Q2}$: We first clarify that the symbols $\mathbf{X}\_{t}^{\overline{\mathrm{Or}}}$ and $\mathbf{\bar{X}}^{\mathrm{Or}}$ are not the same. Their definitions are given in Lines 141 and 291: $\mathbf{X}\_{t}^{\overline{\mathrm{Or}}}$ denotes replacing the original missing entries with zeros, whereas $\mathbf{\bar{X}}^{\mathrm{Or}}$ denotes applying linear interpolation to the original missing data. Also, the notation: $\mathbf{s}$ denotes the true score, $\mathbf{s}\_\theta$ is the score function learned by the neural network (e.g., U-Net), and $\mathbf{s}\_\theta^*$ is the optimal score function learned by the network. Importantly, due to the presence of original missing data, the true score function computed from complete datascore function is never obtained during either training or inference.
>
> In Section 4, we theoretically analyze the bias between the ground truth score function and the score function with original missing data input as $\mathbf{s}(\mathbf{X}\_{t}, t) - \mathbf{s}(\mathbf{X}\_{t}^{\overline{\mathrm{Or}}}, t)$ from the definition of the score function (gradient of the log probability), not from the neural network. Importantly, if the score function is biased due to original missing data, then the neural network can only learn a score estimator $\mathbf{s}_\theta$ that approximates $\mathbf{s}(\mathbf{X}_t^{\overline{\mathrm{Or}}}, t)$ rather than the true score $\mathbf{s}(\mathbf{X}_t, t)$. This mismatch inevitably propagates to the inference stage and leads to biased imputation results. This fundamental difficulty motivates us to apply the latent diffusion to learn the original missing data in an unsupervised way, as shown in Eq. 9, to reduce this bias caused by original missing data, rather than simply replacing them with zeros as done in CSDI.
>
> If linear approximation refer to the Taylor expansion of a function, the equation holds exactly whenever higher-order derivatives is 0, unless a derivative does not exist. As a result, there is no ilnear approximation in the Taylor expansion.
>
> $\textbf{Q3}$: Please refer to our response to Q2.
>
> $\textbf{Q4}$: In principle, our method can be applied to other data types, such as images. However, the neural network architecture would need to be adapted to the specific domain—for example, using convolutional or vision-transformer-based modules for images—to effectively capture the structural characteristics of the data.

---

### Official Review · Reviewer_W2GG · 2025-10-31

**Soundness:** 3
**Presentation:** 3
**Contribution:** 3
**Rating:** 4
**Confidence:** 4

**Summary:**

This paper addresses missing data imputation in multivariate time series (MTS). While recent score-based generative models have shown promise for MTS imputation, most neglect the original missing data during training, leading to biased score estimation.
The authors theoretically analyze this bias within the DDPM framework, demonstrating that ignoring missing patterns distorts the learned score function—even for non-missing data points. To overcome this, they propose HSGM, which combines latent- and observation-space diffusion using the chain rule of probability. Experiments on four benchmark datasets show that HSGM outperforms existing methods in imputation accuracy and uncertainty estimates, while effectively correcting score bias.

**Strengths:**

S1: The paper directly addresses the critical issue of ignoring original missing in score-based imputation.
S2: The paper rigorously analyzes the bias in the score function induced by missing data under the DDPM framework, offering a solid mathematical foundation for why existing methods fail and justifying the need for HSGM.
S3: The paper presents comprehensive experimental results across a variety of benchmark datasets, showcasing HSGM's superior performance compared to existing methods.

**Weaknesses:**

W1: The paper outlines the weaknesses of individual imputation models in isolation, but it does not attempt to group the literature into coherent categories. A taxonomy that first classifies existing approaches and then highlights the shared limitations within each category would provide readers with a clearer understanding.
W2: The Introduction does not clearly explain why the 'original missing effect' warrants attention, nor does it provide a rationale for choosing DDPM as the framework for theoretical analysis.
W3: Several components of the proposed method, such as VAE, CSDI, VPSDE, and cross-attention, are based on existing techniques, which limits the novelty of the method.
W4: The authors should further explore how their method benefits downstream tasks like prediction and classification in the Experiments.

**Questions:**

1.Why was linear interpolation chosen for handling original missing data? When 80 % of the values are fake, would the performance be seriously impacted?
2.The method described in the paper follows a multi-stage process (latent diffusion - VAE decoding - observation diffusion), with each stage feeding its output directly into the next. In this pipeline, is there a risk that errors from earlier stages could accumulate and amplify in later stages? If so, how might this affect the final imputation accuracy, and what measures have been taken to mitigate such error propagation?

---

> ### Author Response · Authors · 2025-11-20
> **Response to Reviewer W2GG  (Part I)**
>
> We thank the reviewer for their valuable insights and the time dedicated to reviewing our paper. We address each of the concerns and suggestions in detail below.
>
> $\textbf{W1}$: Missing data is indeed a pervasive challenge in real-world data collection, and a wide range of imputation techniques have been proposed—spanning traditional statistical approaches, discriminative deep models, and modern generative models. Since our work specifically targets generative imputation with a focus on score-based diffusion models, our discussion concentrated on this subset of the literature rather than providing a full taxonomy across all families of methods.
> For a comprehensive and well-organized taxonomy of multivariate time-series imputation approaches—including statistical, discriminative deep learning, and generative paradigms—we refer readers to an extensive recent survey:
>
> Jun Wang, Wenjie Du, Yiyuan Yang, Linglong Qian, Wei Cao, Keli Zhang, Wenjia Wang, Yuxuan
> Liang, and Qingsong Wen. Deep learning for multivariate time series imputation: A survey. arXiv
> preprint arXiv:2402.04059, 2024.
>
> This survey categorizes existing methods systematically and discusses the common limitations within each group. Our paper builds upon these insights and focuses on addressing the core weaknesses of generative approaches, particularly score-based models.
>
> $\textbf{W2}$: Thank you for the insightful question. Most existing imputation baselines rely on a supervised learning setup, where models are trained on simulated missing data for which ground-truth values are available. In contrast, original missing data—the entries truly absent in the dataset—have no ground truth and are typically treated as being independent of the simulated missing pattern. As a consequence, prior works often default to imputing zeros (or fixed constants) for these original missing entries during training. In many real-world domains, especially healthcare, the proportion of original missing data is substantial. Imputing these entries is considerably more challenging because the true values are inherently unknown. This fundamental difficulty motivates our work: understanding and mitigating the impact of original missing data is crucial for building reliable and unbiased imputation models.
>
> Our theoretical analysis shows that this practice can lead to a biased estimation of the score function when the original missing mechanism is correlated with the simulated one. This bias propagates through the diffusion process and ultimately degrades imputation quality. This motivates our approach: by operating in the latent space and designing a framework that mitigates the influence of unknown original missing values, we can reduce this bias and achieve more accurate imputation. Please refer to our response to Reviewer niE6  regarding comment W1 for more detail.
>
> Regarding the choice of DDPM for theoretical analysis: DDPM-based diffusion models are among the most widely used, stable, and well-understood score-based generative frameworks. In multivariate time-series imputation specifically, nearly all recent diffusion-based methods—including our baselines such as CSDI, FGTI, and Diffputer—are built upon the DDPM formulation. For this reason, grounding our theoretical analysis in the DDPM framework allows us to provide results that are directly comparable and broadly applicable to existing MSTI diffusion models.
>
> $\textbf{W3}$: Thank you for the comment. Our work is centered on the theoretical analysis of bias in the learned score function under practical missing-data conditions, as presented in Section 4. We show that this bias can be significantly reduced in the latent space (Eq. 9), which forms the core contribution of our paper.
>
> Given this focus, we deliberately build on stable and widely adopted components—such as VAE, CSDI, VPSDE, and cross-attention—not to claim architectural novelty, but to ensure a clean and controlled environment for validating our theoretical findings. Using familiar and well-established modules helps isolate the effect of our theoretical insight without confounding it with additional architectural innovations.

---

> ### Author Response · Authors · 2025-11-20
> **Response to Reviewer W2GG (Part II)**
>
> $\textbf{W4}$: Thank you for the insightful suggestion. Our work is primarily focused on imputation performance and on establishing the theoretical foundation of score-function bias, as well as demonstrating how this bias can be effectively reduced. These aspects form the core contribution of the paper. For this reason—and due to space limitations—we did not extend our study to downstream tasks such as prediction or classification.
>
> We fully agree, however, that evaluating the impact of improved imputation on downstream tasks is valuable. Our framework is compatible with standard predictive and classification pipelines, and we expect that reducing score bias and producing higher-quality imputations would translate into performance gains in these settings. Conducting a comprehensive downstream evaluation is an important and meaningful direction for future work, and we appreciate the reviewer for highlighting it.
>
> $\textbf{Q1}$: Linear interpolation is used to handle original missing data because we pretrain the VAE before diffusion. If we simply impute original missing entries with zeros, the input matrix becomes extremely sparse, which significantly degrades the VAE’s ability to learn meaningful representations. Linear interpolation, while still an approximation, preserves basic temporal structure in the multivariate time series (MTS) and provides a more informative and stable input for VAE pretraining.
>
> In datasets with very high missing rates, such as healthcare datasets like P2012 where around 80\% of values are missing, zero imputation can severely distort the data. Instead, our latent diffusion framework learns the underlying latent distribution, allowing the model to infer values closer to the true data distribution. Experiments (Table 3) show that imputing original missing data in this way significantly improves performance, consistent with our theoretical analysis.
>
> $\textbf{Q2}$: Yes, there is a potential risk of error propagation in the multi-stage pipeline, where outliers or randomness from earlier stages could affect the final imputation accuracy. To mitigate the impact of outliers, standard preprocessing steps—used both in our baseline models and in the MIMIC dataset—are applied to remove extreme values. Additionally, during training and evaluation, we monitor performance using ground truth values in the validation dataset to ensure stability.
>
> To reduce the influence of stochasticity in data generation, we sample from the noise 100 times and compute the mean values as input to the next stage, as illustrated in Algorithm 1. In the latent diffusion stage, we further adopt probability flow deterministic sampling, which minimizes randomness in the generative process. Together, these measures help control error accumulation and improve the robustness of the final imputation results.

---

> > ### Author Response · Authors · 2025-11-26
> >
> > Regarding W2, as inspired by Reviewer dwA9, we conducted the related experiments. The details and results are presented in Response to Reviewer dwA9 (Part Ⅲ) of Answer 2.

---

### Meta-Review · Area_Chair_vPFj · 2025-12-26

**Summary:**

This paper proposes a DDPM-based generative imputation method for multivariate time series. The authors theoretically formalize the inherent bias in score estimation when 'original' missing data (lacking ground truth) is ignored during training, demonstrating that this can distort score estimation even for observed data points. To mitigate this, they introduce the Hierarchical Score-based Generative Model (HSGM), which utilizes a VAE with normalizing flows to establish a latent space. The method employs two distinct diffusion models: a latent diffusion module that generates representations of missing values in an unsupervised manner, and an observation-space diffusion module that refines the decoded 'draft' into a final imputation. The authors demonstrate that this hierarchical approach achieves state-of-the-art point-accuracy on several benchmarks, particularly in scenarios with high missing rates.

**Reviewer Concerns:**

Reviewer W2GG

- Lack of coherent surveys on existing methods.
I believe this issue was not fully resolved by the authors, as they ultimately refer to an existing survey. However, I do not consider this weakness significant enough to warrant downgrading the submission.

- Introduction not clear.
The points of confusion were addressed in the rebuttal, but these clarifications were not sufficiently reflected in the revised manuscript.

- Limited novelty.
The authors argued that the primary contribution of the paper lies in its theoretical understanding of the problem rather than in novel model design. However, I do not think the novelty concern has been fully addressed. See further discussion in the “Reviewer concerns” section below.

- Lack of downstream task experiments.
The authors acknowledged this limitation. That said, I believe requesting downstream task evaluations may be beyond the intended scope of this submission.

Reviewer xT6t

- Limited novelty.
See the discussion above.

- Clarification needed for time complexity in comparison to baselines.
This issue was addressed in the rebuttal with a detailed analysis.

- Multi-sample averaging as an alternative option.
This technique is already used in the current experiments and does not fundamentally mitigate the bias issue.

Reviewer niE6

- Propositions 4.1 and 4.2 are not appropriate, as they assume diffusion models trained with full data.
This concern appears to stem from a misunderstanding. The theoretical analysis does not assume access to the full dataset during training. This issue has been resolved.

**Reviewer Scores:**

Overall, the reviewers have partially addressed the minor concerns, including clarity of presentation, correction of misunderstandings in the theoretical statements, and the addition of a complexity analysis.
However, I remain skeptical that the authors’ rebuttal sufficiently resolves the criticism regarding limited novelty. My concern is not simply that the proposed method is an incremental combination of existing techniques. Rather, while the authors argue that the primary contribution lies in the theoretical understanding—specifically, the analysis of potential bias in score estimation under missing data—I find this contribution unconvincing for two reasons.
First, the core theoretical finding is not particularly surprising: while the exact magnitude of the bias may not be predictable, the very existence of bias under missing data is largely expected.
More importantly, the proposed algorithm, HGSM, is not tightly coupled with the theoretical analysis. The theory identifies a potential issue in estimation, but the proposed solution does not appear to be directly derived from or guided by this theory. Instead, it is presented as a heuristic motivated by intuition—namely, that coupling two diffusions in the observed and latent spaces may mitigate the bias. The manuscript does not convincingly establish a link between the theoretical insights and the algorithmic design. Roughly speaking, the proposed method would still make sense even without the theory itself.

---

### Decision · Program_Chairs · 2026-01-26

Reject